# Giant activity-induced elasticity in entangled polymer solutions

Davide Breoni [1,2,7], Christina Kurzthaler [3,4,5], Benno Liebchen [6], Hartmut Löwen [1] & Suvendu Mandal [1,6] ✉

One of the key achievements of equilibrium polymer physics is the prediction of scaling laws governing the viscoelastic properties of entangled polymer systems, validated in both natural polymers, such as DNA, and synthetic polymers, including polyethylene, which form materials like plastics. Recently, focus has shifted to active polymers systems composed of motile units driven far from equilibrium, such as California blackworms, self-propelled biopolymers, and soft robotic grippers. Despite their growing importance, we do not yet understand their viscoelastic properties and universal scaling laws. Here, we use Brownian dynamics simulations to investigate the viscoelastic properties of highly-entangled, flexible self-propelled polymers. Our results demonstrate that activity enhances the elasticity by orders of magnitude due to the emergence of grip forces at entanglement points, leading to its scaling with polymer length $\sim L$. Furthermore, activity fluidizes the suspension, with the long-time viscosity scaling as $\sim L^2$, compared to $\sim L^3$ in passive systems. These insights open new avenues for designing activity-responsive polymeric materials.

Entanglement of polymer-like entities underlies a variety of intricate phenomena across multiple scales that are driven by different active processes, ranging from self-propulsion to growth [Fig. 1a]. Inside living cells the interplay of tread-milling actin filaments and molecular motors serves as a key example of entangled activated polymers that fluidize the cell cytoskeleton and enable cell migration[1–6]. A further example is the packaging of chromatin, consisting of meter-scale DNA, into the micron-sized nucleus, where its enzyme-driven disentanglement is vital for the transcription of genetic material[7–13]. Entangled collective lifeforms represent another class of active polymers that widely appear in microbial settings, where microorganisms grow into physically entangled structures to adapt within their ecological niches[14–16], and at the macroscale, enabling California blackworms to resist and dynamically respond to environmental stresses[17]. In the realm of bio-inspired engineering, active entangled polymers occur in the form of soft robotic grippers that are able to capture objects of

various shapes[18,19], emphasizing the potential of entangled filaments for applications. Therefore, understanding the interplay of entanglement and different forms of activity is not only fundamental to living systems but also crucial for designing and processing new soft materials with tailored mechanical properties. Despite the widespread occurrence of active polymers in nature and technology, not much is known about their characteristic physical properties.

Strongly entangled polymers at thermodynamic equilibrium have been extensively explored using the tube model[20,21]. A major breakthrough lies in the theoretical prediction of elastic properties of entangled linear polymer melts based on their stress autocorrelation function, which exhibits a prominent plateau at intermediate times, characterizing the elastic response, and relaxes exponentially at long times. The relation between phenomenological parameters of the underlying tube model and microscopic system properties to predict the stress plateau has been established by analyzing the polymers'

[1]Institut für Theoretische Physik II: Weiche Materie, Heinrich Heine-Universität Düsseldorf, Universitätsstraße 1, 40225 Düsseldorf, Germany. [2]INFN-TIFPA, Trento Institute for Fundamental Physics and Applications, Via Sommarive 14, I-38123 Trento, Italy. [3]Max Planck Institute for the Physics of Complex Systems, Nöhnitzer Straße 38, 01187 Dresden, Germany. [4]Center for Systems Biology Dresden, Dresden, Germany. [5]Cluster of Excellence, Physics of Life, TU Dresden, Dresden, Germany. [6]Technische Universität Darmstadt, Karolinenplatz 5, 64289 Darmstadt, Germany. [7]Present address: Soft matter and Biophysics Institute, Università di Trento, Via Sommarive 14, I-38123 Trento, Italy. ✉e-mail: mandal@hhu.de; suvendu.mandal@pkm.tu-darmstadt.de

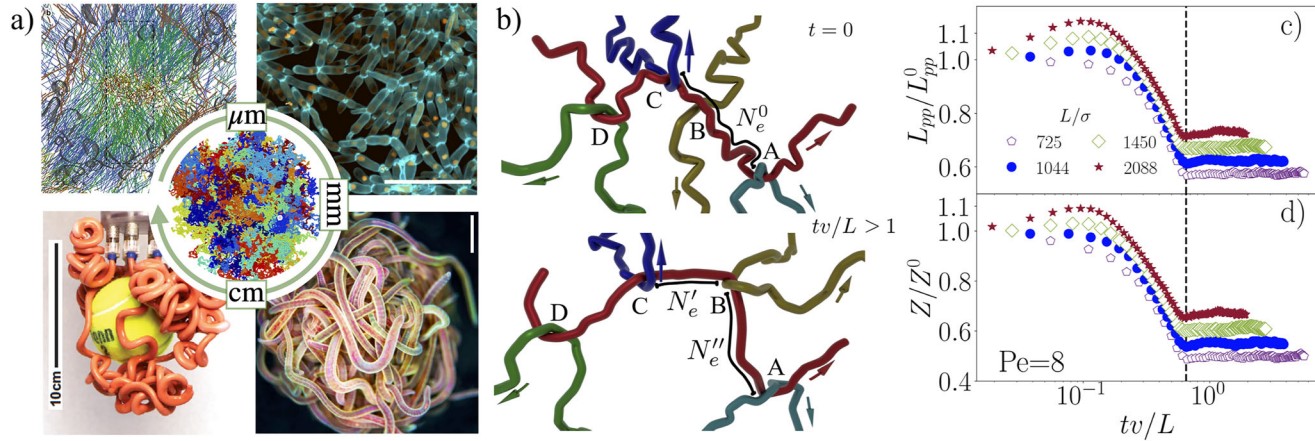

**Fig. 1 | Exemplars of entangled active matter across scales and illustration of the physical mechanism driving their material properties. a** (*Top left panel*) 3D segmentation of actin filaments in native podosomes, scale bar 200 nm. Color code indicates their orientation. From ref. 73. (*Top right panel*) Growing community of snowflake yeast, scale bar 50 μm. From ref. 16. (*Bottom left panel*) Synthetic active filaments change curvature upon pneumatic actuation to entangle and capture different objects. From ref. 19. (*Bottom right panel*) Entangled blob of California blackworms, scale bar 3 mm. From ref. 17. Reprinted with permission from AAAS. At the center a simulation snapshot of entangled, flexible polymers (each polymer has its own color) is shown. **b** 3D illustration depicting the primitive path of a test polymer (red line) confined within an effective tube formed by surrounding self-propelled polymers at various times $t$. In the equilibrium state

($t$ = 0), a combination of strong entanglement points (A, C, and D) and weak entanglement points (B) coexists, with strong entanglements distinguished by the presence of hairpin structures. Due to the activity, before reaching the steady state $t \gtrsim L/v$, the number of strong entanglement points increases (as shown by the yellow polymer wrapping around the red polymer at point B), leading to the elongation of the primitive path. The direction of self-propulsion is indicated by colored arrows, while the distance between successive entanglement points defines the entanglement length $N_e$. **c** Contour length of the primitive path $L_{pp}$, normalized by the equilibrium primitive path $L_{pp}^0$ as a function of time for different polymer lengths $L$ and fixed Péclet number Pe = 8. Time is rescaled by the ratio of polymer length to self-propulsion velocity $L/v$. **d** Number of entanglement points $Z$, normalized by the number of entanglement points $Z^0$ for Pe = 0, as a function time.

primitive paths[22,23], corresponding to the axes of entangled polymer tubes. While the stress plateau of linear polymer solutions remains unaffected by external driving, tuning the topological properties of polymers can lead to a qualitative change in the stress relaxation dynamics[24]. Active polymer models have successfully captured complex biological processes, from the chiral migration of malaria parasites[25] and the collective organization of cyanobacteria[26], to the treadmilling of filaments driving bacterial cell division[27], but the viscoelastic properties of such active systems remain largely unexplored. We propose that incorporating active components in addition to tuning the entanglement has the potential to reveal new exciting physics, yet, no universal behaviors or scaling predictions have been established so far to guide experimental progress.

Here, we employ Brownian dynamics simulations to characterize the viscoelastic properties of a minimal model of self-propelled entangled polymers. Our results reveal a remarkable amplification of the stress plateau - an effect intricately linked to the interplay of active motion and entanglement, leading to the emergence of grip forces. In particular, neighboring polymers form hairpin structures that exert forces, pulling a test polymer in the direction of their self-propulsion, effectively preventing its sliding at the entanglement points. Importantly, the magnitude of these grip forces depends on the self-propulsion velocity. Subsequently, we demonstrate that the stress autocorrelation functions for a broad range of polymer lengths and Péclet numbers can be collapsed onto a single master curve by identifying the characteristic activity-induced energy and disengagement time. Finally, we predict that the activity fluidizes the system which displays a long-time viscosity scaling with the square of the polymer length.

## Results
### Model: active entangled polymer simulations
We perform 3D Brownian dynamics simulations of highly-entangled polymer solutions of $N$ self-propelled, flexible polymer chains using the bead-spring model[28]. Each chain consists of $N_p$ monomers with diameter $\sigma$ and has a length of $L = N_p \sigma$. The connectivity and repulsion

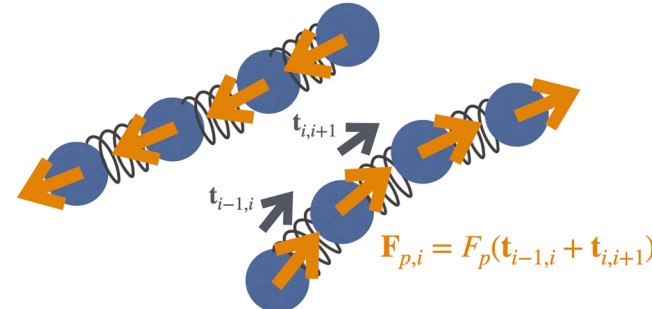

**Fig. 2 | Representative examples of active polymers.** Each polymer is self-propelled along its tangent vector, $(\mathbf{t}_{i-1,i} + \mathbf{t}_{i,i+1})$.

of the beads are modeled using the finitely extensible nonlinear elastic potential (FENE)[28] and the Weeks-Chandler-Andersen potential (WCA)[29] with energies $\epsilon_{\text{FENE}}$ and $\epsilon_{\text{WCA}}$, respectively. Angular interactions along chain backbones are captured using a bending potential for each monomer $U_{\text{ang},i} = \kappa \sum_{j=i-1,i,i+1}(1 - \mathbf{t}_j \cdot \mathbf{t}_{j+1})$, where $\mathbf{t}_j = (\mathbf{r}_{j+1} - \mathbf{r}_j)/(|\mathbf{r}_{j+1} - \mathbf{r}_j|)$ represents the tangent vector between consecutive monomers having positions $\mathbf{r}_j$ and $\kappa$ corresponds to the bending energy. The polymers are subject to Brownian motion modeled by stochastic forces $\mathbf{F}_{r,i}$, where $\langle F_{r,i}^\alpha(t)F_{r,j}^\beta(t')\rangle = 2k_B T\zeta\delta_{ij}\delta_{\alpha\beta}\delta(t'-t)$ with friction coefficient $\zeta$ and thermal energy $k_B T$. In the overdamped regime, momentum dissipates immediately, ensuring that the system consistently remains at the bath temperature, $k_B T$, making an explicit thermostat unnecessary[30]. Their self-propulsion is modeled by an active force $\mathbf{F}_{p,i} = F_p(\mathbf{t}_{i-1,i} + \mathbf{t}_{i,i+1})$ acting tangentially to the polymer contour[31–37], so that (without interactions) each bead moves at a velocity of $v = |\mathbf{F}_{p,i}|/\zeta$ ($|\mathbf{F}_{p,i}|$ being constant across all monomers) [see Fig. 2]. Thus, the equation of motion for each monomer reads

$$\zeta\frac{d\mathbf{r}_i}{dt} = -\nabla_i U + \mathbf{F}_{p,i} + \mathbf{F}_{r,i}. \tag{1}$$

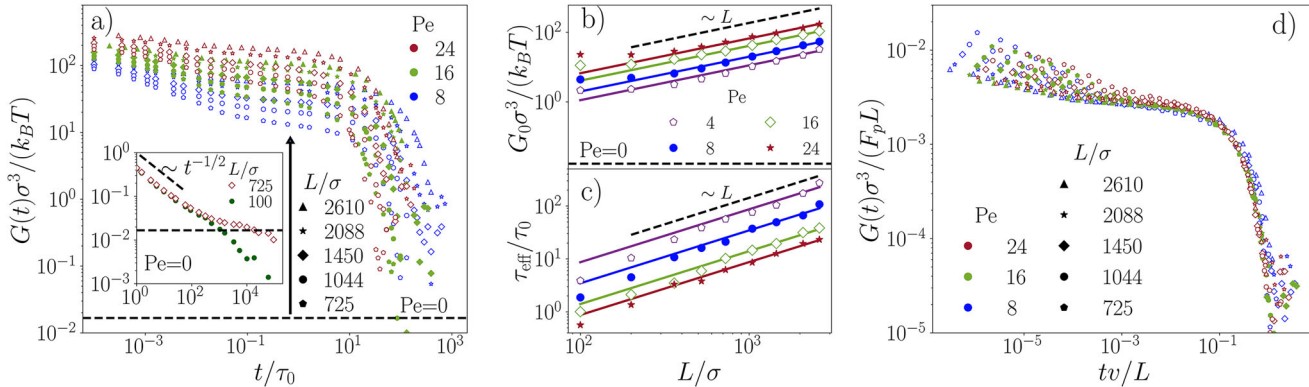

**Fig. 3 | Impact of activity on the stress autocorrelation function. a** Stress autocorrelation function $G(t)$ for different Péclet numbers and polymer lengths $L$ as a function of time. Inset: The stress autocorrelation function in equilibrium for $L = 725\sigma$ validates the well-established prediction $G_0 \simeq 4\rho k_B T/(5N_e^0)$ (dashed line), where $N_e^0$ is the entanglement length between two subsequent entanglement points. Axes are labeled as in the main figure. Black arrow indicates the giant activity-induced stress plateau compared to the equilibrium state. **b** Stress plateau $G_0$ and (**c**) disengagement time $\tau_{\text{eff}}$ as a function of polymer length $L$ extracted from our simulations for a wide range of Péclet numbers. **d** Rescaled stress autocorrelation function $G(t)\sigma^3/F_p L$ as a function of the rescaled time $tv/L$.

Dimensionless parameters, derived from length and time units ($\sigma$ and $\tau_0 = \sigma^2/D_0$, with $D_0 = k_B T/\zeta$ as the short-time diffusion coefficient of a monomer), include the Péclet number (Pe $= v\sigma/D_0$) for assessing the significance of active motion relative to diffusion, along with coupling parameters ($\epsilon_{\text{WCA}}/k_B T$, $\epsilon_{\text{FENE}}/k_B T$, and $\kappa/k_B T$). In our simulations, the friction coefficient is set to $\xi = k_B T/D_0 = 1$. For the FENE potential, we define the maximum bond length as $R_0 = 1.5\sigma$. Additionally, we define the dimensionless density $\rho^\star = N_{\text{tot}}\sigma^3/V$, where $V$ denotes the volume of the simulation box. We keep fixed values of $\rho^\star = 0.85$, $\epsilon_{\text{WCA}}/k_B T = 1.0$, $\epsilon_{\text{FENE}}/k_B T = 30$, and $\kappa/k_B T = 1.0$, while systematically varying the polymer length ($L/\sigma = 100, \ldots 2610$), resulting in a dimensionless entanglement length $N_e \cong 41$[38]. Equations of motion are solved numerically using a modified version of LAMMPS with a time step of $\delta t = 10^{-4}\tau_0$. Equilibration is achieved through a bond-swapping algorithm with core softening [see Supplementary Information (SI) and refs. 39–41], and all time measurements are referenced from this equilibration point. Notably, both active and passive highly-entangled polymer systems exhibit an ideal chain scaling relation for the end-to-end distance $R_{ee} \propto L^{1/2}$, in contrast to dilute active polymer solutions[32], indicating that activity does not affect this scaling [see SI]. We further determined quantities, such as the entanglement time $\tau_e \simeq N_e^2 \simeq 1600\tau_0$, representing the time required for polymers to experience topological constraints, and the Rouse time $\tau_R \simeq L^2$, reflecting the longest relaxation time of the polymer chains within their confining tube, analogous to previous studies[42].

## Activity-enhanced elasticity

The viscoelastic properties of polymer solutions are encoded in the stress autocorrelation function

$$G(t) = \frac{V}{3k_B T}\sum_{\alpha \neq \beta}\left\langle \sigma_{\alpha\beta}(t)\sigma_{\alpha\beta}(0)\right\rangle, \qquad (2)$$

where the sum runs over all off-diagonal components of the stress tensor $\sigma_{\alpha\beta}$ (i.e., $\sigma_{xy}$, $\sigma_{yz}$, $\sigma_{xz}$), and $\langle \ldots \rangle$ denotes the ensemble average, evaluated in the steady state ($t > L/v$). Specifically, the off-diagonal stress tensor components are calculated as $\sigma_{\alpha\beta} = -1/(2V)\sum_{k=1}^{N_p N}\sum_{l=1}^{N_p N} F_{kl}^{\alpha} r_{kl}^{\beta}$, where $N_p$ is the number of beads per polymer, $N$ is the number of polymer chains, $F_{kl}^{\alpha}$ represents the $\alpha$ component of the force between the $k$th and $l$th beads, and $r_{kl}^{\beta}$ is the $\beta$ component of their distance[12,30]. In equilibrium systems, for the case of short, unentangled linear polymer solutions, this yields the power-law dynamics described by the Rouse model, $G(t) \sim t^{-1/2}$[43]. In contrast, highly-entangled polymers are forced to move along the

direction of their contour, while their motion perpendicular to it is restricted to a tube-like region formed by the surrounding polymers [Fig. 1b]. Consequently, the stress autocorrelation function exhibits a plateau $G_0$ at intermediate times and an exponential decay $G_0 e^{-t/\tau_{\text{eff}}}$ at long times. The stress plateau $G_0$, a hallmark of entangled polymer chains, quantifies the elasticity of the system, while the disengagement time $\tau_{\text{eff}} \sim L^3$ in equilibrium corresponds to the characteristic time the polymer requires to move its own length $L$ along the tube.

To investigate the effect of activity, we compute the stress autocorrelation function $G(t)$ for self-propelled polymers of different lengths, $L/\sigma = 100, \ldots 2610$, and Péclet numbers, Pe $= 1, \ldots 24$, see Fig. 3a. At very short times ($t \lesssim 10^{-3}\tau_0$), the active polymer solution is slightly harder ($G(t)$ increases by a factor of 4 compared to the passive counterpart [see SI]), which can be attributed to the increased fluctuations exhibited by the self-propelled polymers within their tubes.

At intermediate times, $t \sim \tau_0$, the difference between the stress autocorrelation function $G(t)$ of passive and active systems becomes significantly larger, which becomes apparent in an increase of the plateau height $G_0$ by three orders in magnitude [see Fig. 3a]. This amplification arises from grip forces exerted on the red test polymer by neighboring polymers [Fig. 1b]. First, these neighboring polymers form hairpin structures around the test polymer, stretching its primitive path, thereby slowing down the relaxation of $G(t)$ as the test polymer traverses within an elongated tube. This effect becomes pronounced when we keep the Péclet number constant while increasing the polymer length [Fig. 3b]. Second and more strikingly, these grip forces also act as barriers, effectively preventing the test polymer from sliding at the entanglement points. This results in a substantial increase in the plateau height as Pe increases at a fixed polymer length [Fig. 3a and SI]. Hence, both mechanisms contribute to a giant enhancement of the elastic stress plateau height, a phenomenon exclusive to self-propelled entangled systems. When the test polymer disengages from its tube, the grip forces imposed by the surrounding polymers diminish, leading to a relaxation of the stress autocorrelation function from the plateau at long times [see Fig. 3a].

This physical picture can be corroborated by measuring the average contour length of the primitive path, denoted as $L_{pp}$, and the average number $Z$ of entanglement points. To elucidate topological entanglement dynamics, we employed the Z1+ topological analysis algorithm[38,44–47], which systematically undergoes a sequence of geometric minimizations. The primitive path is rigorously defined as the shortest path between the two ends of a polymer chain while preserving its topological uncrossability. At intermediate times $tv/L \sim 0.1$, our simulations show that upon increasing the polymer length at a fixed Pe = 8, $L_{pp}$

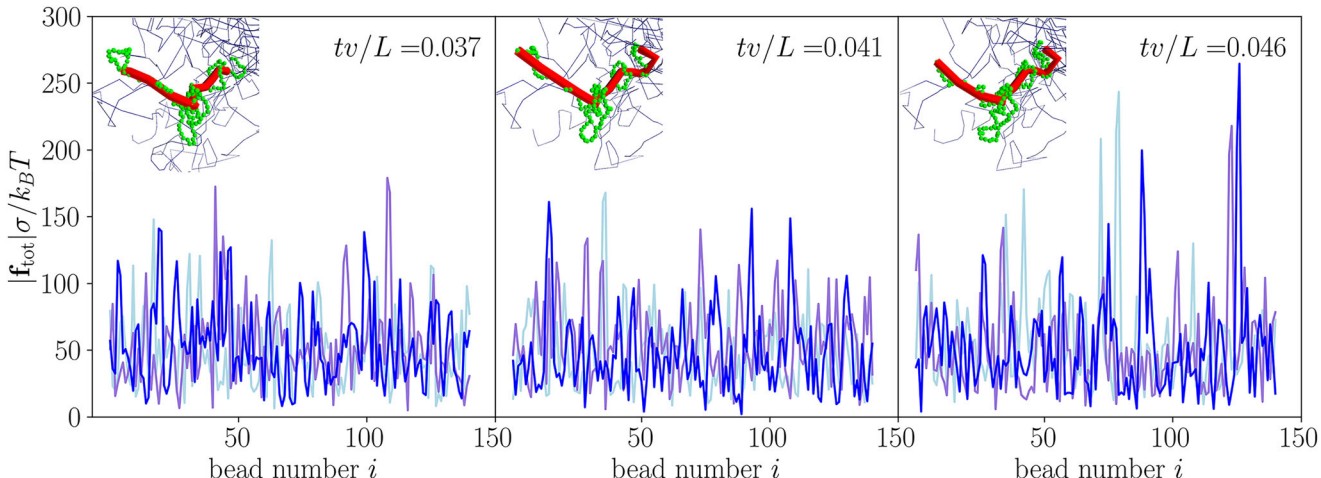

**Fig. 4 | Time evolution of grip forces.** Total force on the first 150 beads of three test polymer chains as a function of bead number, showing the intensification of grip forces over time: $tv/L = 0.037$, $tv/L = 0.041$, and $tv/L = 0.046$. The polymer has a length of $L = 1450\sigma$ and a Péclet number of Pe = 4. As time progresses, the grip forces exerted by neighboring polymers become stronger, leading to the stretching of the test polymer at key entanglement points. The insets depict one of the test polymers (green) and its primitive path (red), along with the primitive paths of neighboring polymers (blue), emphasizing the evolving topological interactions and constraints over time. Over the observed time interval, the contour length of the test polymer increases from $377\sigma$ to $389\sigma$.

and $Z$ increase by 10% compared to the passive counterpart [see Fig. 1c, d]. This observation suggests that the active system becomes more entangled, with the number of entanglement points rising from $Z = 105$ to 115 for $L/\sigma = 2088$. Moreover, we evaluated the entanglement length $N_e$ using the relation $N_e = (N_p - 1)\langle R_{ee}^2 \rangle / \langle L_{pp}^2 \rangle$[38,46]. In contrast to $L_{pp}$, the end-to-end distance $R_{ee}$ exhibits a gradual decrease until it eventually saturates at long times ($tv/L \gg 1$) at a fixed Pe = 8 [see SI]. Consequently, at intermediate times ($tv/L \sim 0.1$), we observe a reduction of approximately 30% in $N_e$ relative to the passive counterpart [see SI].

To investigate the nature of grip forces, we extended our analysis by quantifying the forces along the polymer contour. In Fig. 4, we present data for a subsection of three test polymers (first 150 beads), showing the total force on each bead, $|\mathbf{f}_{tot}| = |\mathbf{F}_{WCA} + \mathbf{F}_{FENE} + \mathbf{F}_{ang} + \mathbf{F}_p|$, over time. Here, $\mathbf{F}_{WCA}$, $\mathbf{F}_{FENE}$, and $\mathbf{F}_{ang}$ represent the forces due to the WCA, FENE, and angular potentials, respectively. At $tv/L = 0.037$, the total force fluctuates around $|\mathbf{f}_{tot}| \approx 50 k_B T/\sigma$. As time progresses, grip forces from neighboring polymers (blue) intensify, increasing $|\mathbf{f}_{tot}|$ by a factor of 4 at $tv/L = 0.046$. These amplified forces stretch the primitive path (red) at key entanglement points, resulting in an increase in the contour length of the test polymer from $377\sigma$ to $389\sigma$.

It is tempting to validate the giant increase in the stress plateau $G_0$ via the well-established relation for equilibrium systems $G_0 \simeq 4\rho k_B T/(5N_e)$[48], which remains constant at a fixed density and temperature, as the tube diameter ($\sim \sqrt{N_e}$) remains unaltered with increasing polymer length. However, our observations reveal a 30% decrease in $N_e$ with increasing polymer length $L$ at a fixed Péclet number (Pe = 8), while the stress plateau $G_0$ increases by orders of magnitude. By employing a dimensional argument, we show that the enhanced stress plateau can rather be related to the active energy of a single polymer $F_p L$, where $F_p$ denotes the magnitude of the active force. For large Pe $\gg 1$, this energy dominates over thermal energy and thus represents the relevant energy scale of our system, leading to our prediction $G_0 \sim F_p L/\sigma^3$. To quantify this phenomenon, we show the plateau height $G_0$ as a function of the polymer length $L$ for a range of Péclet numbers in Fig. 3b. It turns out that $G_0$ indeed increases linearly as a function of the polymer length in the highly entangled regime ($L \gtrsim 350\sigma$) in striking contrast to its passive counterparts. This occurs because the polymers are forced to move within elongated tubes as well as the system gets highly entangled (the number of entanglement points $Z$ increases compared to the passive counterpart). However, for unentangled chains with $L \lesssim 100\sigma$, the stress plateau vanishes and we recover an algebraic decay $\sim t^{-1/2}$, in agreement

with the Rouse model, which validates the idea that the stress plateau is a unique feature of highly entangled polymer solutions (see SI).

At long times $t \gg \tau_0$, the stress autocorrelation function follows the expected exponential decay $G(t) \sim G_0 \exp(-t/\tau_{eff})$, where $\tau_{eff}$ represents the disengagement time of our active system [see Fig. 3a]. At these times, the transverse motion becomes nearly frozen, allowing the polymer to self-propel and diffuse freely along the tube at time-scales of $L/v$ and $\sim L^3$, respectively. The disengagement time $\tau_{eff}$ is determined by the faster of these two mechanisms and we use the interpolation formula given below as an estimate:

$$\tau_{eff}^{-1} = D_0 \sigma/L^3 + v/L. \quad (3)$$

Remarkably, our computer simulations show that active entangled polymers relax much faster than their passive counterparts, resulting in a disengagement time that scales as $\tau_{eff} \sim L$ [see Fig. 3c]. This is in contrast to the passive case, where the disengagement time scales as $\sim L^3$ for larger polymer lengths, as observed in experiments[49]. By combining the relevant time $\tau_{eff} \sim L/v$ and energy scales $G_0 \sim F_p L/\sigma^3$, the data collapse onto a single curve at intermediate and long times, as depicted in Fig. 3d. The data collapse is excellent over nearly three decades in time, confirming our predictions.

To further characterize the viscoelastic properties of active entangled polymers, we compute the frequency-dependent storage $G'(\omega)$ and loss moduli $G''(\omega)$ by applying a Fourier transform to the stress relaxation modulus $G(t)$[43]. Our results reveal a well-defined crossover frequency $\omega_c$, where $G'(\omega)$ and $G''(\omega)$ intersect, marking the transition from a viscous-dominated to an elastic-dominated regime. Notably, this crossover frequency scales as $\omega_c \sim 1/\tau_{eff}$, confirming that relaxation dynamics are governed by activity-driven disengagement [see SI].

Additionally, we compare the active timescale, $\tau_{eff} = L/v$, to the entanglement time, $\tau_e \sim N_e^2$, which represents the time required for a polymer chain to encounter topological constraints from entanglements in equilibrium. Our analysis indicates that $\tau_{eff}$ is typically shorter than $\tau_e$ (i.e., $\tau_{eff} \lesssim \tau_e$). Furthermore, in the regime where $\tau_{eff} \gg \tau_e$ (i.e., Pe $\ll 1$), the polymers remain in the linear response regime, behaving similarly to equilibrium systems despite the influence of active forces. This comparison highlights the interplay between activity and entanglement effects, deepening our understanding of the dynamics that extend beyond those of passive polymers.

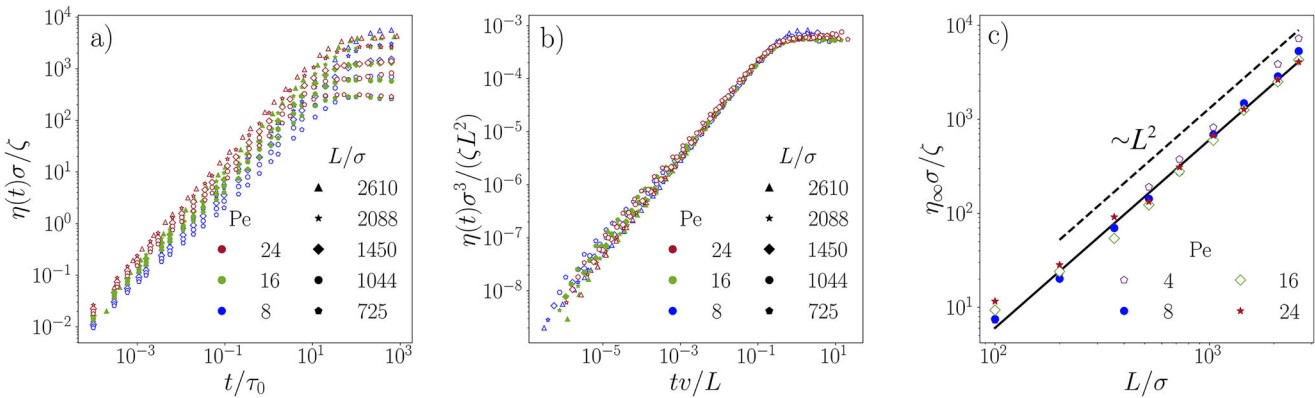

**Fig. 5 | Fluidization of active solutions compared to their passive counterparts.** **a** Time-dependent viscosity for a wide range of Péclet numbers and polymer lengths. **b** A data collapse is obtained by rescaling the viscosity by $\eta(t)\sigma^3/\zeta L^2$ and the time scale by $tv/L$. **c** Long-time viscosity $\eta_\infty$ as a function of polymer length $L$ extracted from simulations for a wide range of Péclet numbers. The black line indicates the scaling of $\eta_\infty \sim L^2$.

## Time-dependent viscosity

Following our previous predictions ($G_0 \sim LF_p/\sigma^3$ and $\tau_{\rm eff} \sim L/v$), the viscosity is expected to scale as $\eta \sim G_0\tau_{\rm eff} \sim L^2$ [see Methods]. Only recently, it has been claimed that in the hydrodynamic limit (i.e., at long times and at large length scales) the Green-Kubo relation is valid even for suspensions of active dumbells[50]. Our verification of key conditions, including the statistical independence of thermal and active forces, the stability of the steady state, and the isotropy of the system, and confirmation that spatial velocity correlations decay sufficiently fast, ensures the applicability of the formalism also for our work [see SI]. Thus, the Green-Kubo relation offers access to the time-dependent viscosity of our entangled system via

$$\eta(t) = \int_0^t G(t')\,{\rm dt'}, \qquad (4)$$

which is shown in Fig. 5a over 6 decades in time. Our study suggests that at short times $t \lesssim \tau_0$, activity and entanglement play a minor role, but at intermediate times the data become significantly different. Rescaling the data accordingly ($\eta(t)/L^2$ and $tv/L$), we find a collapse onto a single master curve over 4 orders of magnitude in time [Fig. 5b].

Finally, as our data saturate at long times we can estimate the stationary viscosity of the system via $\eta_\infty \equiv \lim_{t\to\infty} \eta(t)$. The predicted scaling $\eta_\infty \sim L^2$ is confirmed by an asymptotic data collapse in the regime of high entanglement ($L/\sigma \gtrsim 350$) and high Péclet numbers (Pe $\gtrsim 8$) [Fig. 5c]. Hence, highly-engtangled active solutions follow a generic scaling of $\sim L^2$, which is distinct from the characteristic $L^3$ scaling that broadly applies to equilibrium systems. Deviations become apparent for shorter polymer lengths ($L/\sigma \leq 250$), where the solution becomes less entangled. This can be attributed to the fact that as we increase the Péclet number, the tube diameter ($\sim \sqrt{N_e}\sigma$)[43] also becomes larger. Therefore, it requires even longer polymers to observe a highly-entangled state.

## Discussion

Our work represents a minimal model for understanding the elasticity of active entangled polymer solutions, revealing a novel scaling law for elasticity, $G_0 \sim L$, for a fixed Péclet number. This contrasts with the conventional passive entangled polymer solutions, where elasticity remains constant regardless of polymer lengths[22]. The activity-enhanced elasticity elucidated in our study is a critical property for collective life forms, which could provide individuals resistance to environmental stresses[16,17,51–55] via tuning their own activity. This intricate feature also lays the foundation for numerous technological applications, involving new materials composed, for

example, of microscale activated nanotubes[56], synthetic polymer chains[57], rigid helical filaments[58,59], or soft shape-changing actuators[18,19]. Furthermore, our findings hold significant promise in predicting the elasticity of growing entangled living systems, such as bacterial colonies[14,16], as our simulation results across various polymer lengths can effectively map onto growing matter. In particular, we anticipate that the plateau of the stress autocorrelation function, and hence the elasticity, increases over time due to the growth of filamentous bacterial strands. Looking ahead, it would be of great interest to investigate how these active entangled polymer solutions behave under deformation or shear[60–64], introducing another time-scale (inverse shear rate) that could reveal nontrivial viscoelastic properties. Additionally, a direct validation of our scaling predictions could be achieved by further experimental studies[51] utilizing controlled worm lengths.

While our study focuses on self-propelled flexible polymers, many polymers found in nature are semiflexible[65–70]. Therefore, a future challenge is to include the finite bending rigidity of polymers in our analysis and explore how elasticity and disengagement time vary with swimming speed. Anticipating the occurrence of grip forces in any active entangled systems, our minimal model offers the potential to establish a universal understanding of biological filament behavior and contribute to the development of advanced materials with tailored viscoelastic properties, including synthetic cells[71,72].

Finally, real active polymer systems, such as actin filament networks[65], involve additional complexities, including dynamic cross-linkers that bind and detach from filaments. These interactions could influence the structural and mechanical properties of the network. Thus, future work should incorporate such dynamic crosslinking effects, along with more detailed interactions, into our model to bridge the gap between minimalistic simulations and experimental systems.

## Methods
### Zero-shear viscosity
The zero-shear viscosity is expressed as:

$$\eta_\infty = \int_0^\infty G(t)\,{\rm d}t \simeq \int_0^{\tau_{\rm relax}} G(t)\,{\rm d}t \simeq G_0\tau_{\rm relax},$$

where $G(t)$ is the stress relaxation modulus. This integral is dominated by the terminal relaxation time $\tau_{\rm relax} \sim L^3$ in passive systems, leading to the approximation $\eta_\infty \sim G_0\tau_{\rm relax}$. In passive systems, $G_0$ is constant and proportional to $k_BT$ per entanglement, thus yielding the familiar scaling $\eta_\infty \sim L^3$ as predicted by de Gennes[21].

In contrast, our study reveals that in active systems, $G_0$ scales with the input active energy, $F_pL \sim vL$. For Pe $\gtrsim 1$, the terminal relaxation

time is set by active motion, $\tau_{\text{relax}} \sim \tau_{\text{eff}} \sim L/v$. Thus, the zero-shear viscosity for active systems becomes:

$$\eta_\infty = \int_0^\infty G(t)\,\mathrm{d}t \simeq \int_0^{\tau_{\text{eff}}} G(t)\,\mathrm{d}t$$
$$\sim G_0 \tau_{\text{eff}} \sim F_p L \tau_{\text{eff}} \sim (vL)(L/v) \sim L^2.$$

This scaling represents a fundamental shift from passive systems, where $\eta_\infty \sim L^3$, highlighting the different dissipation mechanisms. In active systems, fluidization occurs, lowering viscosity as entanglements diminish, even for long polymers ($L/\sigma = 2610$).

## Movie

The movie (Supplementary Movie 1) illustrates the dynamic evolution of primitive paths involving a test polymer (in red) along with its neighboring polymers (in blue) in a simulation setting characterized by Pe = 4, $L/\sigma = 1450$, and $\rho^\star = 0.85$. Notably, it reveals an increase in the primitive path, expanding from $L_{pp}/\sigma = 291.3$ to $L_{pp}/\sigma = 348.6$ at intermediate times $tv/L \sim 0.14$. Ultimately, the contour length of the primitive path $L_{pp}$ decreases by 40% compared to its passive counterpart at long times ($tv/L \geq 1$).

## Data availability

The data are available from the corresponding author upon request.

## Code availability

The computer code used for simulations is available in the Code Ocean repository [DOI 10.24433/CO.3999799.v1].

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

## Acknowledgements

S.M. gratefully acknowledges Robert S. Hoy and Joseph D. Dietz for valuable discussions regarding the equilibration of entangled polymer solutions, as well as Martin Kröger for sharing the topological analysis code before publishing it. The work of D.B. was supported within the EU MSCA-ITN ActiveMatter (proposal No. 812780). H.L. acknowledges funds from the German Research Foundation (DFG) project LO 418/29-1.

## Author contributions

S.M. designed the project. D.B. and S.M. performed the simulations and analyzed the data. D.B., C.K., B.L., H.L., and S.M. discussed the results and implications. S.M. and C.K. wrote the manuscript with the help of all other authors.

## Funding

## Competing interests

The authors declare no competing interests.
