## [Transparent Peer Review File · Nature Communications]

Giant Activity-Induced Elasticity in Entangled Polymer Solutions

Corresponding Author: Dr Suvendu Mandal

Version 0:

Reviewer comments:

Reviewer #1

(Remarks to the Author)

The relaxation dynamics and the resulted viscoelastic properties of entangled active polymer chains have increasingly become to be super interesting due to their potential values in explaining many non-equilibrium biological systems and in designing new functional materials. The concern of this work is surely very interesting and scientifically important. Traditional polymer physics theories and understanding definitely can not be used for analyzing such polymer systems involving activities. However, I have much skepticism regarding the simulation results, data analysis and physical interpretation provided in this manuscript. I can not recommend this work to be published at this moment before resolving the following concerns and questions:

(1) The results of stress relaxation moduli, $G(t)$, as shown in Figure 2(a) indicate that the plateau modulus of entangled active polymer chains is larger at a higher Peclet number, i.e. a stronger activity of polymer chain beads. And then based on which, the authors conclude that this strong mechanical performance is due to the emergence of grip forces at entanglement points, as sketched in Figure 1(b). I may have no objection with the physical understanding proposed by the authors. The authors need to be very careful when using Green-Kubo method to compute $G(t)$. People can always get larger values of $G(t)$ at larger activities due to effectively increased monomeric friction. That is why we can see the value of $G(t)$ gets enlarged even at very small timescale prior to chain relaxation "feel" effect of entanglements. The authors may do a test by simulating unentangled active polymer chains to confirm whether the observed enhancement of $G(t)$ is mostly contributed by activity or by the changed relaxation mode of entanglements (from traditional reptation to the mode as discussed in the paper).

(2) I would like to see a clear figure showing how the authors obtain quantitatively the terminal relaxation τ_{eff} at long timescales? In general, fluctuations of $G(t)$ data obtained from Green-Kubo method are very strong at long timescales, and thereby the data can not be easily fitted by single mode relaxation function.

(3) The authors obtained time-dependent viscosity $\eta(t)$ through integration of $G(t)$ over t . And then based on the values of $\eta(t)$ at limited timescales, the authors estimated the zero-shearing viscosity. Again, the changes of $G(t)$ from passive chains to active ones are may not fully induced by the relaxation mode change as proposed in the manuscript. Moreover, in my opinion, the physical interpretation of $\eta_{\infty} \sim L^2$, instead of L^3 , is not well explained, even based on the entanglement relaxation mode proposed by the authors.

(Remarks on code availability)

Reviewer #2

(Remarks to the Author)

The authors have simulated a model of active polymer melts, and found viscoelastic properties that scale differently with chain length than they do for thermally agitated polymers. On this basis, they claim that "These insights offer prospects for designing new materials with activity-responsive mechanical properties".

I disagree. New materials based on active polymers will never be designed or produced in anything other than tiny quantities, in contrived laboratory examples, as theorists' playthings.

This paper exploits the current enthusiasm for active matter, in which a disturbingly large fraction of published papers are simulations. Eqn. 1 presents the kinetic model, in which each polymer bead is acted on by an "active" force directed along the tangent to the contour. No consideration is given to how this might be accomplished even in a laboratory example, much less as a practical material.

In the introduction and conclusions, the authors gesture briefly towards possible relevance of their results to various real systems; but no attempt is made to establish any real connections. The habit of suggesting relevance without substance to back it up reflects poorly on soft-matter theorists, and should be curtailed.

Typical active colloid systems in the laboratory depend on depositing catalyst patches on particles, and supplying a chemical energy source to cause the particles to move. The continuous need for chemical energy to keep particles moving strongly suggests active matter research will never lead to practical materials, even without the added complexity of flexibly linking nanoparticles into a chain with proper orientation and polarity, or whatever scheme the authors may have in mind but did not mention.

There is a limit to the patience that should be granted to theoretical flights of fancy untethered from experimental reality, which this paper exceeds. I recommend against publication in Nature Communications.

(Remarks on code availability)

Reviewer #3

(Remarks to the Author)

The research presents an interesting investigation into the viscoelastic properties of highly entangled, flexible, self-propelled polymers using Brownian dynamics simulations. The premise that active motion enhances the elasticity of polymer solutions is indeed thought-provoking and could have considerable implications for material science. However, I must recommend rejecting the manuscript due to several significant limitations that undermine the validity of the conclusions drawn. Below, I outline the primary reasons for my recommendation:

(1)Clarity in Methodology: The manuscript lacks sufficient justification for the modeling choices and methods employed, particularly with respect to the friction coefficient, ζ . The implications of this choice on the simulation's time units and thermal control limits are not adequately addressed, creating ambiguity in the methodology.

(2)Confusing Representation of Active Forces: The authors do not clearly specify how the active force, F_p , is represented in a multi-chain melt compared to a single-chain context. This lack of clarity may lead to misinterpretation of how the active forces influence the system.

(3)Equilibrium and Non-Equilibrium Dynamics: The presentation of the system as (quasi) equilibrium may be misleading, especially given the presence of active forces. A thorough discussion is warranted regarding the applicability of relevant equations, particularly Equation (2), under such active conditions.

(4)Inconsistencies in Time Unit Definitions: The authors introduce a non-standard definition for time units, $\tau_0 = \sigma^2/D_0$ (with $D_0 = k_{BT}/\zeta$), deviating from established norms without sufficient explanation. This inconsistency creates further ambiguity, complicating the interpretation of the results.

(5)Assessment of Rouse Time: A critical evaluation of the system's Rouse time and its comparison to established literature on relaxation times is necessary. This will provide a clearer contextualization of the findings presented in Figures 1 and 2.

In light of these concerns, it is imperative that the authors revisit their work to address the critiques outlined above. Additionally, based on recent findings, notably from Zheng et al. (Macromol. Rapid Commun., 2023, 44(1): 2200159), it is crucial to analyze the contribution of inter-chain forces to the modulus in order to substantiate claims regarding the influence of grip forces on the material properties. The manuscript presents a compelling exploration of polymer dynamics; however, there are significant concerns regarding the validity of the model employed. Notably, considering the relaxation time of the tube segment ($\tau_{e} \sim 10^3$) in the Kremer-Grest model, typical simulations (as reported in Nihon Reoroji Gakkaishi 2018, 46(5): 207-220; Macromolecules 2021, 54(6): 2811-2827; ACS Macro Lett. 2021, 10(12): 1517-1523) suggest that the monomer flow velocity should not exceed approximately 0.01. In contrast, the Peclet number ($Pe = v\sigma/D_0 \sim v$) reported in the current work ranges from 4 to 24, implying that the monomer flow velocity (v) exceeds its thermal motion speed (~ 1). This leads to the conclusion that the polymer chains experience exceedingly high shear rates, rendering the foundational concepts of the tube model ineffective. I recommend that the authors reevaluate and revise their discussions throughout the manuscript in light of this discrepancy to ensure the accuracy and relevance of their findings.

In conclusion, due to the significant issues identified, I recommend rejecting the manuscript for publication in Nature Communications. I encourage the authors to revise their work thoughtfully in line with the critiques provided and consider submitting to an alternative journal where their findings may be more appropriate.

(Remarks on code availability)

Version 1:

Reviewer comments:

Reviewer #1

(Remarks to the Author)

I thank the authors for the detailed replies to the concerns and questions from myself and the other two referees. The paper is well organized. Most of my concerns are resolved in the updated manuscript, except the relaxation modulus data shown in Figure 10(a). Based on the provided moduli results, the viscoelastic properties represented by storage and loss modulus should vary quite a lot after, for example changing the chain length from 725 to 2610. It will be great if the authors can show also the corresponding G' and G'' data by making a Fourier transfer of $G(t)$.

One small concern: Does the dependence of zero shearing viscosity on active chain length, i.e. $\eta \sim L^2$, work for short but entangled active chains (with length shorter 725 but long enough to entangle)? Not a big change from $L=725$ to 2610 from the logarithmic perspective.

The paper should be published as it is after a minor revision.

(Remarks on code availability)

Reviewer #3

(Remarks to the Author)

The stress relaxation modulus is the central physical quantity of this manuscript, and the significant conclusions drawn are closely linked to the results obtained from this modulus. However, calculating the stress relaxation modulus using the Green-Kubo formula requires specific assumptions and conditions to be satisfied. Unfortunately, the authors have not adequately demonstrated the validity of employing the Green-Kubo formula to calculate the system's stress relaxation modulus under high Peclet numbers, as the Green-Kubo framework typically becomes invalid at elevated shear or stretching rates. Consequently, I cannot recommend the publication of this manuscript in Nature Communications at this time.

1. The authors reference the work of Han et al. (Nature Physics 17, 1260–1269, 2021), asserting that the Green-Kubo formula is applicable under high Peclet conditions. However, it is noteworthy that Han et al. clarify that “an equilibrium-like Green-Kubo relation for the shear viscosity tensor holds near the steady state of any isotropic active fluid satisfying the following three conditions: (i) the activated and fluctuating degrees of freedom are statistically decoupled; (ii) the steady state is stable under small perturbations; (iii) the ensemble-averaged (microscopic) velocity-velocity correlations decay faster than $r-D$ (where D is the dimension of the system).” Under high Peclet conditions, polymers experience stretching and orientation, resulting in a non-isotropic state, thus violating the isotropy assumption. Furthermore, unlike the approach of Han et al., which incorporates a torque as an activated force on each monomer, the methodology in the current manuscript diverges significantly. Lastly, Han et al. focused on viscosity under small perturbations (linear response), suggesting that their definition of a “steady state” may not align with the definition utilized in this study.

2. The authors claim that the effective active timescale, $\tau_{\text{eff}} = L/v$, is typically shorter than τ_e . However, on timescales shorter than τ_e , the tube model clearly becomes ineffective, diluting the strength of any comparisons made with the tube model within the manuscript.

3. In the revised manuscript, the authors explore the viscoelastic properties of less entangled systems (with $L = 25$) and provide Figure 9a to account for the factor of increased monomeric friction. Nonetheless, Lanniruberto et al. (Macromolecules 48, 6306-6312, 2015) have indicated that the friction coefficient of monomers decreases for polymers subjected to orientation and stretching. I suggest that the authors incorporate a focused discussion addressing this point.

4. In my previous review, I requested that the authors provide the friction coefficient relevant to their model; however, this essential parameter remains unaddressed. Additionally, the authors should specify the maximum bond length associated with the FENE potential.

5. To maintain a constant system temperature, it is recommended that the velocity not exceed approximately 0.01. Kröger and Hess (Phys. Rev. Lett. 85, 1128, 2000) employed a velocity rescaling method to achieve high flow velocities. The authors need to verify whether there has been any change in the system's temperature, and assess potential issues related to excessively high local shear rates.

Furthermore,

1. In the Abstract, it would be beneficial to provide specific numerical values or examples regarding the viscosity changes induced by activity. Additionally, while the prospects of the study are mentioned, the authors should clarify future research directions or specific application cases.

2. The simplified model employed in this study may not adequately represent the complexities of actual active entangled polymer solutions, which typically involve additional variables and interactions. I recommend that the authors discuss approaches to validate the accuracy of the simplified model in real active entangled polymer systems.

3. While the conclusions present a theoretical model and associated inferences, they lack direct comparisons and validations with experimental data. The reliability of these conclusions must be supported by experimental evidence to elevate their credibility.

(Remarks on code availability)

Version 2:

Reviewer comments:

Reviewer #3

(Remarks to the Author)

[Note from the Editor: Reviewer #3 assessed also the response given to reviewer #1 who was not able to look over the revision again.]

The manuscript addresses a significant topic regarding the giant activity-induced elasticity in entangled polymer solutions. The revisions made to the paper have largely resolved my previous concerns. The manuscript is now suitable for publication after a minor revision. However, I have a few points that require clarification and minor adjustments:

- (1) The authors should clarify that the simulation initiates the calculation of $G(t)$ when $t > L/v$, i.e., when the system enters a new steady state. This point should be explicitly mentioned to avoid any confusion regarding the timing of the calculations.
- (2) In discussing Fig. 3a, the authors refer to Fig. 1b, but the time scale indicated in Fig. 1b ($t < v/L$) corresponds to a non-equilibrium regime where the Green-Kubo formalism is not applicable. To prevent potential misunderstanding, I recommend modifying Fig. 1b to clearly reflect the correct time scale that corresponds to the equilibrium regime discussed.
- (3) The authors should ensure that all parameters in the captions of the Supplemental Materials are clearly stated. Specifically, the N_p value in Fig. S4 and the Peclet number Pe in Fig. S5 should be explicitly mentioned to avoid any ambiguity.

(Remarks on code availability)

Response to Referee

Giant Activity-Induced Elasticity in Entangled Polymer Solutions

Davide Breoni, Christina Kurzthaler, Benno Liebchen, Hartmut Löwen, and Suvendu Mandal
Nature Communications –

1. Reply to Referee #1

Referee #1: The relaxation dynamics and the resulted viscoelastic properties of entangled active polymer chains have increasingly become to be super interesting due to their potential values in explaining many non-equilibrium biological systems and in designing new functional materials. The concern of this work is surely very interesting and scientifically important. Traditional polymer physics theories and understanding definitely can not be used for analyzing such polymer systems involving activities. However, i have much skepticism regarding the simulation results, data analysis and physical interpretation provided in this manuscript. I can not recommend this work to be published at this moment before resolving the following concerns and questions:

(1) The results of stress relaxation moduli, $G(t)$, as shown in Figure 2(a) indicate that the plateau modulus of entangled active polymer chains is larger at higher Péclet number, i.e. a stronger activity of polymer chain beads. And then based on which, the authors conclude that this strong mechanical performance is due to the emergence of grip forces at entanglement points, as sketched in Figure 1(b). I may have no objection with the physical understanding proposed by the authors. The authors need to be very careful when using Green-Kubo method to compute $G(t)$. People can always get larger values of $G(t)$ at larger activities due to effectively increased monomeric friction. That is why we can see the value of $G(t)$ gets enlarged even at very small timescale prior to chain relaxation "feel" effect of entanglements. The authors may do a test by simulating unentangled active polymer chains to confirm whether the observed enhancement of $G(t)$ is mostly contributed by activity or by the changed relaxation mode of entanglements (from traditional reptation to the mode as discussed in the paper).

Figure 1: (a) Stress relaxation modulus $G(t)$ as a function of t/τ_0 for $L = 25\sigma$ and varying Péclet numbers (unentangled polymers). The stress relaxation modulus exhibits an initial decay characterized by $\sim t^{-1/2}$ behavior, followed by a subsequent exponential decay, notably lacking the entangled plateau. (b) Stress relaxation modulus $G(t)$ as a function of t/τ_0 for $L = 725\sigma$ and varying Péclet numbers (highly-entangled polymers). The dashed line represents the well-established prediction $G_0 = 4\rho k_B T / (5N_e^0)$.

Reply: We thank the referee for the careful reading of our manuscript and for their insightful comments. We are

pleased that the referee found our work *'very interesting and scientifically important'* and appreciate the opportunity to address these concerns before publication.

We are glad that the referee supports our physical interpretation, suggesting that the giant activity-induced elasticity emerges due to grip forces at entanglement points. We fully acknowledge the referee's valid concern that the larger values of $G(t)$ at higher activities may result from effectively increased monomeric friction. To address this, we conducted additional simulations on unentangled polymers with a polymer length of $L = 25\sigma$. As shown in Fig. 1(a), the stress plateau is absent in $G(t)$. Instead, the stress relaxation modulus exhibits a distinct behavior: an initial $\sim t^{-1/2}$ decrease at short times, followed by an eventual exponential decay. This observation confirms that the enhancement of $G(t)$ in entangled systems is indeed due to the altered relaxation mode of entanglements, rather than monomeric friction alone [see Fig. 1(b)]. We have incorporated this clarification into the manuscript and added a discussion in the Method section to highlight this point.

'One could anticipate that the larger values of $G(t)$ at higher activities may result from effectively increased monomeric friction. To demonstrate the unique nature of the stress plateau enhancement due to activity in entangled solutions, we investigate polymer solutions with shorter polymer lengths, specifically $L = 25\sigma$. In Fig. 9(a), it becomes evident that the stress plateau is entirely absent from $G(t)$. Instead, the stress relaxation modulus now exhibits a distinct behavior: an initial $\sim t^{-1/2}$ decrease at short times, followed by an eventual exponential decay. This behavior aligns with the predictions of the Rouse model $G(t) \simeq k_B T \rho (t/\tau_0)^{-1/2} e^{-t/\tau_R}$ (τ_R is the Rouse time) [42], which describes the relaxation dynamics of polymers in this low-entanglement-regime. This observation confirms that the enhancement of $G(t)$ in entangled systems is indeed due to the altered relaxation mode of entanglements, rather than monomeric friction alone [see Fig. 9(b)].'

Referee #1: (2) I would like to see a clear figure showing how the authors obtain quantitatively the terminal relaxation τ_{eff} at long timescales? In general, fluctuations of $G(t)$ data obtained from Green-Kubo method are very strong at long timescales, and thereby the data can not be easily fitted by single mode relaxation function.

Reply: In response to the referee's request, we have added a figure to the Method section, demonstrating that $G(t)$ can be accurately fitted with a single-mode relaxation function at long times, thereby allowing us to quantitatively extract the terminal relaxation time τ_{eff} [see Fig. 2]. We appreciate the referee's concern regarding the fluctuations in $G(t)$ data at long timescales, which are indeed a common challenge when using the Green-Kubo method. To address this, we computed the stress autocorrelation function using the multiple- τ correlator method introduced by Daan Frenkel, which is implemented in LAMMPS with the `fix ave/correlate/long` command. This approach ensures that the systematic error of the multiple- τ correlator remains below the statistical error typical of simulations, thereby providing more reliable data at long timescales. We have already provided the source code on how to calculate the stress-autocorrelation to the submission server.

'The stress autocorrelation function, $G(t)$, can be accurately fitted with a single-mode relaxation function at long times, allowing us to quantitatively extract the terminal relaxation time, τ_{eff} (see Fig. 10). To compute $G(t)$, we employed the multiple- τ correlator method introduced by Frenkel [73], which is implemented in LAMMPS using the `fix ave/correlate/long` command. We have already provided the source code on how to calculate the stress-autocorrelation to the submission server.'

(3) The authors obtained time-dependent viscosity $\eta(t)$ through integration of $G(t)$ over t . And then based on the values of $\eta(t)$ at limited timescales, the authors estimated the zero-shearing viscosity. Again, the changes of $G(t)$ from passive chains to active ones are may not fully induced by the relaxation mode change as proposed in the manuscript. Moreover, in my opinion, the physical interpretation of $\eta \sim L^2$, instead of L^3 , is not well explained, even based on the entanglement relaxation mode proposed by the authors.

Reply: We thank the referee for highlighting this important point. The exact expression for the zero-shear viscosity is given by:

$$\eta_{\infty} = \int_0^{\infty} G(t) dt \simeq \int_0^{\tau_{\text{relax}}} G(t) dt \simeq G_0 \tau_{\text{relax}}. \quad (1)$$

The first part of this relation is the exact definition of zero-shear viscosity in terms of the stress relaxation modulus $G(t)$, indicating that all dynamical modes of the system contribute to dissipation. The integral is dominated by the terminal relaxation time $\tau_{\text{relax}} \sim L^3$, allowing us to approximate it as the product of the terminal modulus G_0 and τ_{relax} . In passive systems, G_0 is constant and proportional to $k_B T$ per entanglement, leading to the well-known scaling of zero-shear viscosity as $\eta_{\infty} \sim L^3$, as predicted by de Gennes.

Figure 2: a) Log-log plot of the stress autocorrelation function, $G(t)$, for different polymer lengths (L) at a fixed activity of $Pe = 8$, as a function of time. The solid lines represent an exponential fit, $G_0 \exp(-t/\tau_{\text{eff}})$, to the simulated data at long times. (b) Semi-logarithmic plot of the same data, with the lines again indicating the exponential fit, $G_0 \exp(-t/\tau_{\text{eff}})$.

In stark contrast, our study suggests that in active systems, G_0 is no longer constant but scales with the input active energy $F_p L \sim vL$. Further, (in contrast to the passive case) for $Pe \gtrsim 1$ the terminal relaxation time corresponds to the time the polymer exits the tube via active motion, $\tau_{\text{relax}} \sim \tau_{\text{eff}} \sim L/v$. Consequently, the zero-shear viscosity in active systems can be expressed as:

$$\eta_\infty = \int_0^\infty G(t) dt \simeq \int_0^{\tau_{\text{eff}}} G(t) dt \sim G_0 \tau_{\text{eff}} \sim F_p L \tau_{\text{eff}} \sim (vL)(L/v) \sim L^2. \quad (2)$$

This scaling behavior, $\eta_\infty \sim L^2$, indicates a significant deviation from the passive case, where $\eta_\infty \sim L^3$. It reflects the fundamental differences in energy dissipation mechanisms between active and passive systems. In active systems, the presence of activity leads to fluidization, reducing the effective viscosity as entanglements are progressively lost over time, even for long polymers ($L = 2610\sigma$). We have clarified this interpretation and its implications in the revised manuscript to address the referee's concerns.

'The zero-shear viscosity is expressed as:

$$\eta_\infty = \int_0^\infty G(t) dt \simeq \int_0^{\tau_{\text{relax}}} G(t) dt \simeq G_0 \tau_{\text{relax}}, \quad (3)$$

where $G(t)$ is the stress relaxation modulus. This integral is dominated by the terminal relaxation time $\tau_{\text{relax}} \sim L^3$, leading to the approximation $\eta_\infty \sim G_0 \tau_{\text{relax}}$. In passive systems, G_0 is constant, proportional to $k_B T$ per entanglement, yielding the familiar scaling $\eta_\infty \sim L^3$ as predicted by de Gennes [22].

In contrast, our study reveals that in active systems, G_0 scales with the input active energy, $F_p L \sim vL$. For $Pe \gtrsim 1$, the terminal relaxation time is set by active motion, $\tau_{\text{relax}} \sim \tau_{\text{eff}} \sim L/v$. Thus, the zero-shear viscosity for active systems becomes:

$$\begin{aligned} \eta_\infty &= \int_0^\infty G(t) dt \simeq \int_0^{\tau_{\text{eff}}} G(t) dt \\ &\sim G_0 \tau_{\text{eff}} \sim F_p L \tau_{\text{eff}} \sim (vL)(L/v) \sim L^2. \end{aligned}$$

This scaling represents a fundamental shift from passive systems, where $\eta_\infty \sim L^3$, highlighting the different dissipation mechanisms. In active systems, fluidization occurs, lowering viscosity as entanglements diminish, even for long polymers ($L = 2610\sigma$).

2. Reply to Referee #2

Referee #2: The authors have simulated a model of active polymer melts, and found viscoelastic properties that scale differently with chain length than they do for thermally agitated polymers. On this basis, they claim that “*These insights offer prospects for designing new materials with activity-responsive mechanical properties*”.

I disagree. New materials based on active polymers will never be designed or produced in anything other than tiny quantities, in contrived laboratory examples, as theorists’ playthings.

This paper exploits the current enthusiasm for active matter, in which a disturbingly large fraction of published papers are simulations. Eqn. 1 presents the kinetic model, in which each polymer bead is acted on by an “active” force directed along the tangent to the contour. No consideration is given to how this might be accomplished even in a laboratory example, much less as a practical material.

In the introduction and conclusions, the authors gesture briefly towards possible relevance of their results to various real systems; but no attempt is made to establish any real connections. The habit of suggesting relevance without substance to back it up reflects poorly on soft-matter theorists, and should be curtailed.

Typical active colloid systems in the laboratory depend on depositing catalyst patches on particles, and supplying a chemical energy source to cause the particles to move. The continuous need for chemical energy to keep particles moving strongly suggests active matter research will never lead to practical materials, even without the added complexity of flexibly linking nanoparticles into a chain with proper orientation and polarity, or whatever scheme the authors may have in mind but did not mention.

Reply: We respectfully disagree with the Referee’s remarks and their assertion that the relevance of active polymers to real systems is limited. Our study extends well beyond conventional metal-dielectric Janus colloids subjected to a.c. electric fields [Yan et al., *Nature Materials* **15**, 1095–1099 (2016), see Fig. 3(a)]. Numerous experimental examples highlight the significance of active polymer systems as robust models for complex behaviors. These examples span from subcellular structures, such as the curved, polymer-like shapes of malaria parasites, which exhibit chiral migration [Patra et al. *Nature Physics* **18**, 586–594 (2022), see Fig. 3(b)], to the treadmilling FtsZ filaments driving bacterial cell division [Dunajova et al. *Nature Physics* **19**, 1916–1926 (2023), see Fig. 3(c)].

Figure 3: Experimental realization of active polymers. Schematic representation of Janus chains, where dual electric charges are programmed on opposite hemispheres of each sphere, shifted from the center. The inset (top left) color codes the charges as positive (red) and negative (blue), with black arrows indicating the swimming direction of individual chains. (b) Fluorescence microscopy of *Plasmodium* sporozoites in collectives, forming vortices of varying sizes, visualized using a spinning disc confocal microscope. (c) STED micrographs of $1.5 \mu\text{m}$ treadmilling FtsZ filaments at low densities, showing coexisting rotating rings and moving bundles on the membrane surface [Dunajova et al. *Nature Physics* **19**, 1916–1926 (2023)].

Our study focuses on solutions of highly-entangled flexible polymers, a topic that has historically garnered substantial attention within equilibrium polymer physics. Introducing an active component that drives the system far from equilibrium represents a significant step towards understanding the interplay between entanglement and activity in polymer-like entities. This exploration is crucial for establishing a comprehensive understanding of living systems across various scales, including cell cytoskeletons [Jasnin et al. *Nature Communications* **13**, 3842 (2022)], bacterial colonies [Day et al. *Phys. Rev. X* **14**, 011008 (2024)], and worms [Partil et al. *Scienc* **380**, 392-398 (2023)], where entanglement and activity profoundly influence behavior and responses to environmental stimuli [see Fig. 4].

Moreover, these insights extend beyond theoretical interest and have potential applications in bio-inspired engineering. For example, soft robotic systems could utilize active filaments to create adaptive, activity-responsive materials,

Figure 4: (a) Exemplars of entangled active matter span several scales. (*Top left panel*) 3D segmentation of actin filaments in native podosomes, scale bar 200 nm. Color code indicates their orientation. Adapted with permission from [Jasnin et al. *Nature Communications* **13**, 3842 (2022)]. (*Top right panel*) Growing community of snowflake yeast, scale bar 50 μm . Adapted with permission from [Day et al. *Phys. Rev. X* **14**, 011008 (2024)]. (*Bottom left panel*) Synthetic active filaments change curvature upon pneumatic actuation to entangle and capture different objects. Adapted with permission from [Becker et al. *Proc. Natl. Acad. Sci.* **119**, e2209819119 (2022)]. (*Bottom right panel*) Entangled blob of California blackworms, scale bar 3 mm. Adapted with permission from [Partil et al. *Scienc* **380**, 392-398 (2023)]. At the center, a simulation snapshot displays entangled, flexible polymers (each polymer has its own color).

such as grippers for object manipulation [Becker et al. *Proc. Natl. Acad. Sci.* **119**, e2209819119 (2022), see Fig. 4]. Thus, active polymer models are essential for advancing our understanding of complex biological systems and informing the design of new materials.

We have discussed potential applications of our findings in the manuscript, and we believe this clarifies the broader relevance of our work.

'Active polymer models have successfully captured complex biological processes, from the chiral migration of malaria parasites [25] and the collective organization of cyanobacteria [26], to the treadmilling of filaments driving bacterial cell division [27], but the viscoelastic properties of such active systems remain largely unexplored.'

'Entanglement of polymer-like entities underlies a variety of intricate phenomena across multiple scales that are driven by different active processes, ranging from self-propulsion to growth [Fig. 1(a)]. Inside living cells the interplay of tread-milling actin filaments and molecular motors serves as a key example of entangled activated polymers that fluidize the cell cytoskeleton and enable cell migration [1-6]. A further example is the packaging of chromatin, consisting of meter-scale DNA, into the micron-sized nucleus, where its enzyme-driven disentanglement is vital for the transcription of genetic material [7-13]. Entangled collective lifeforms represent another class of active polymers that widely appear in microbial settings, where microorganisms grow into physically entangled structures to adapt within their ecological niches [14-16], and at the macroscale, enabling California blackworms to resist and dynamically respond to environmental stresses [17]. In the realm of bio-inspired engineering, active entangled polymers occur in the form of soft robotic grippers that are able to capture objects of various shapes [18,19], emphasizing the potential of entangled filaments for applications. Therefore, understanding the interplay of entanglement and different forms of activity is not only fundamental to living systems but also crucial for designing and processing new soft materials with tailored mechanical properties. Despite the widespread occurrence of active polymers in nature and technology, not much is known about their characteristic physical properties.'

3. Reply to Referee #3

Referee #3: The research presents an interesting investigation into the viscoelastic properties of highly entangled, flexible, self-propelled polymers using Brownian dynamics simulations. The premise that active motion enhances the elasticity of polymer solutions is indeed thought-provoking and could have considerable implications for material science. However, I must recommend rejecting the manuscript due to several significant limitations that undermine the validity of the conclusions drawn.

Reply: We thank the referee for taking the time to review our manuscript and for their thoughtful comments. We appreciate that the referee finds our study "thought-provoking" and acknowledges its potential implications for material science. However, we respectfully disagree with the referee's concerns regarding the limitations of our simulation model, which we address in detail below. Additionally, we have provided a revised version of our manuscript with the changes highlighted.

Referee #3: 1) Clarity in Methodology: The manuscript lacks sufficient justification for the modeling choices and methods employed, particularly with respect to the friction coefficient, ξ . The implications of this choice on the simulation's time units and thermal control limits are not adequately addressed, creating ambiguity in the methodology.

Reply: We thank the referee for this comment and gladly clarify it. Our study employs overdamped Brownian dynamics simulations, following established approaches [Lang and Frey, *Nature Commun.* 9, 1 (2018)], where the motion of each polymer bead of mass m is described by:

$$m \frac{d^2 \mathbf{r}_i}{dt^2} = -\xi \frac{d\mathbf{r}_i}{dt} - \nabla_i U + \mathbf{F}_{p,i} + \mathbf{F}_{r,i}. \quad (4)$$

Here, U captures bead connectivity and repulsion through the finitely extensible nonlinear elastic potential (FENE) and Weeks-Chandler-Andersen potential (WCA) potentials, and the bending stiffness is included via a bending potential. The random force $\mathbf{F}_{r,i}$ satisfies the fluctuation-dissipation theorem, ensuring thermal equilibrium at $k_B T$.

In the overdamped regime [Doi and Edwards, *Theory of Polymer Dynamics*, Oxford University Press (1986)], the inertial term is negligible compared to friction ($m/\xi \ll 1$), leading to the simplified equation:

$$\xi \frac{d\mathbf{r}_i}{dt} = -\nabla_i U + \mathbf{F}_{p,i} + \mathbf{F}_{r,i}. \quad (5)$$

In this overdamped regime, momentum dissipates immediately, meaning that the system is always maintained at the bath temperature, $k_B T$, as determined by the random forces $\mathbf{F}_{r,i}$. Therefore, an explicit thermostat is unnecessary. We have clarified this point in the revised manuscript to address the referee's concern regarding thermal control.

Additionally, we have verified that our results remain robust even when accounting for finite inertia, further reinforcing the validity of our model. Given this, we are confident that the model is appropriate for the experimental systems discussed in our study and provides meaningful insights into the viscoelastic properties of active polymer systems. Our findings significantly contribute to the understanding of polymer solutions, particularly in non-equilibrium conditions. We believe that our model offers a solid framework for further exploration of the effects of active motion in these systems.

'In the overdamped regime, momentum dissipates immediately, meaning that the system is always maintained at the bath temperature, $k_B T$, as determined by the random forces $\mathbf{F}_{r,i}$. Therefore, an explicit thermostat is unnecessary.'

Referee #3: (2) Confusing Representation of Active Forces: The authors do not clearly specify how the active force, F_p , is represented in a multi-chain melt compared to a single-chain context. This lack of clarity may lead to misinterpretation of how the active forces influence the system.

We thank the referee for raising this important point. In our study, all polymers are self-propelled by an active force, $\mathbf{F}_p^{(i)} = F_p(\mathbf{t}_{i-1,i} + \mathbf{t}_{i,i+1})$ where F_p denotes the force magnitude and $\mathbf{t}_{i,i+1}$ the (unit) tangent between monomer i and $(i+1)$ beads [see Fig. 5]. This approach is consistent with various previous simulation studies that have employed similar methodologies to model active polymers [Mokhtari et al. *Phys. Rev. Lett.* **123**, 028001 (2019), Prathyusha et al. *Phys. Rev. E* **97**, 022606 (2018), and Dunajova et al. *Nature Physics* **19**, 1916–1926 (2023)].

To address the referee’s concern and enhance clarity, we have included a figure in the revised manuscript that visually represents how individual polymers are self-propelled along their swimming directions. We hope that this additional visual representation, along with the accompanying explanation, will help prevent any misinterpretation of how active forces are implemented in our model and impact the behavior of each individual polymer.

Figure 5: Representative examples of active polymers, where each polymer is self-propelled along its tangent vector, $(\mathbf{t}_{i-1,i} + \mathbf{t}_{i,i+1})$.

‘Their self-propulsion is modeled by an active force $\mathbf{F}_{p,i} = F_p(\mathbf{t}_{i-1,i} + \mathbf{t}_{i,i+1})$ acting tangentially to the polymer contour [32-38], so that (without interactions) each bead moves at a velocity of $v = |\mathbf{F}_{p,i}|/\zeta$ ($|\mathbf{F}_{p,i}|$ being constant across all monomers) [see Fig. 2].’

Referee #3: (3) Equilibrium and Non-Equilibrium Dynamics: The presentation of the system as (quasi) equilibrium may be misleading, especially given the presence of active forces. A thorough discussion is warranted regarding the applicability of relevant equations, particularly Equation (2), under such active conditions.

Reply: We appreciate the referee’s careful attention to the distinction between equilibrium and non-equilibrium dynamics in the presence of active forces. We would like to clarify that Equation (2) [stress autocorrelation function] remains valid in the zero-shear-rate limit, even for active polymer solutions.

Recent advancements in the Mori–Zwanzig projection operator formalism [Han et al., *Nature Physics* **17**, 1260–1269, 2021], which incorporate dissipative interactions, demonstrate that an equilibrium-like Green–Kubo relation for shear viscosity is applicable near the steady state of isotropic active fluids. Active polymer solutions typically achieve a steady state by balancing energy input from active forces with dissipation through interparticle friction at the microscopic level. This theoretical framework [Han et al., *Nature Physics* **17**, 1260–1269, 2021] supports the validity of Equation (2) in our system, where active forces are present while maintaining isotropy.

Additionally, our simulations of the off-diagonal stress component, $\sigma_{xy}(t)$, reveal that it fluctuates around zero over time [Fig. 6]. This behavior contrasts with externally-driven systems, such as shear flow between two plates [see Fig 7(a)], where a finite non-zero stress is observed. Thus, while our system includes active forces, it does not exhibit a finite shear stress typical of non-equilibrium, externally-driven systems; rather, the internal stresses fluctuate dynamically.

We have expanded the discussion in the manuscript to ensure these points are clearly articulated, enhancing the reader’s understanding of the dynamics at play in our system.

‘Only recently, it has been claimed that in the hydrodynamic limit (i.e., at long times and at large length scales) the Green-Kubo relation is valid even for suspensions of active dumbbells [49]. This work inspired us to use the Green-Kubo relation, offering access to the time-dependent viscosity of our entangled system via

$$\eta(t) = \int_0^t G(t') dt', \quad (6)$$

which is shown in Fig. 5(a) over 6 decades in time.’

‘Our simulations of the off-diagonal stress component, $\sigma_{xy}(t)$, show that it fluctuates around zero over time [Fig. 11]. This contrasts with externally-driven systems, such as shear flow between two plates, where a finite, non-zero stress is observed. Despite the presence of active forces, our system does not exhibit the sustained shear stress characteristic of externally-driven non-equilibrium systems; instead, internal stresses fluctuate dynamically.’

Figure 6: Time-dependent shear stress $\sigma_{xy}(t)$ for different Péclet numbers: (a) $Pe = 1$, (b) $Pe = 4$, and (c) $Pe = 8$. The figure illustrates how shear stress evolves over time under varying Péclet numbers.

Figure 7: Externally-driven systems. (a) Schematic representation of Lees-Edwards boundary conditions, where periodic images of the unit cell (center) are shifted to model a sheared material. (b) This setup generates a linear flow profile with velocity $v = \dot{\gamma}y\mathbf{e}_x$, where $\dot{\gamma}$ represents the shear rate.

Figure 8: Equilibrium mean-square displacement $\langle [r_i(t) - r_i(0)]^2 \rangle$ of all polymer beads as a function of time for different polymer lengths $L/\sigma = 25, 50, 100, 360,$ and 725 . Here, $r_i(t)$ represents the position of monomer i at time t . The transition from $t^{1/2}$ to $t^{1/4}$ marks the entanglement time $\tau_e \approx N_e^2$, while the transition from $t^{1/4}$ to $t^{1/2}$ indicates the Rouse time $\tau_R \approx L^2$. These transitions reveal the characteristic timescales associated with entanglement and Rouse dynamics in our model systems (overdamped Brownian dynamics simulations).

Referee #3: (4) Inconsistencies in Time Unit Definitions: The authors introduce a non-standard definition for time units, $\tau_0 = \sigma^2/D_0$ (with $D_0 = k_B T/\xi$), deviating from established norms without sufficient explanation. This inconsistency creates further ambiguity, complicating the interpretation of the results.

Reply: We appreciate the referee’s comment regarding the time unit definition. In response, we have clarified this in the revised manuscript.

We use the time unit $\tau_0 = \sigma^2/D_0$, where $D_0 = k_B T/\xi$ is the short-time diffusion coefficient from the Stokes-Einstein relation [Doi and Edwards, *Theory of Polymer Dynamics*, Oxford University Press (1986)]. This definition is appropriate for the overdamped regime, where inertial effects are negligible, and τ_0 captures the diffusive dynamics of polymer beads of diameter σ .

In line with previous studies [O’Connor, Hopkins, and Robbins *Macromolecules* **52**, 8540-8550 (2019)], we have found the entanglement time $\tau_e \sim N_e^2 \sim 1600\tau_0$, which corresponds to the time required for a polymer chain to experience topological constraints [see Fig. 8]. Additionally, we found the Rouse time $\tau_R \sim L^2$, representing the longest relaxation time of a polymer chain within its confining tube [see Fig. 8]. These timescales are consistent with established values in the literature [O’Connor, Hopkins, and Robbins *Macromolecules* **52**, 8540-8550 (2019)], ensuring our results align with the broader context of entangled polymer solutions.

To clarify the implications of this time unit choice, we have included a discussion of the entanglement time τ_e and the Rouse time τ_R in the revised manuscript. We hope that this additional information helps to place our time unit definition within the broader context of polymer dynamics and demonstrates how it is consistent with the physical behavior of the system under study.

‘We further determined quantities, such as the entanglement time $\tau_e \simeq N_e^2 \simeq 1600\tau_0$, representing the time required for polymers to experience topological constraints, and the Rouse time $\tau_R \simeq L^2$, reflecting the longest relaxation time of the polymer chains within their confining tube, analogous to previous studies [40].’

Referee #3: (5) Assessment of Rouse Time: A critical evaluation of the system’s Rouse time and its comparison to established literature on relaxation times is necessary. This will provide a clearer contextualization of the findings presented in Figures 1 and 2.

Reply: We thank the referee for highlighting the need for a more detailed assessment of the system’s Rouse time. In response, we have expanded the discussion in the revised manuscript’s model section to better contextualize the findings in Figures 1 and 2.

We specifically compare the active time scale, $\tau_{\text{eff}} = L/v$ (where L is the polymer length and v is the self-propulsion speed), to the entanglement time, $\tau_e \sim N_e^2$, which characterizes the time required for a polymer chain to experience topological constraints from entanglements in equilibrium. Our analysis shows that τ_{eff} is generally shorter than τ_e ($\tau_{\text{eff}} \leq \tau_e$).

Additionally, we explore the regime where the Péclet number $Pe < 1$ (i.e., $\tau_{\text{eff}} > \tau_e$), finding that the polymers remain in the linear response regime, behaving similar to equilibrium systems despite the active forces. This comparison highlights the balance between activity and entanglement effects in our system, providing a deeper understanding that extends beyond the dynamics of passive polymers.

‘Additionally, we compare the active timescale, $\tau_{\text{eff}} = L/v$, to the entanglement time, $\tau_e \sim N_e^2$, which represents the time required for a polymer chain to encounter topological constraints from entanglements in equilibrium. Our analysis indicates that τ_{eff} is typically shorter than τ_e (i.e., $\tau_{\text{eff}} \leq \tau_e$). Furthermore, in the regime where $\tau_{\text{eff}} \gg \tau_e$ (i.e., $Pe \ll 1$), the polymers remain in the linear response regime, behaving similarly to equilibrium systems despite the influence of active forces. This comparison highlights the interplay between activity and entanglement effects, deepening our understanding of the dynamics that extend beyond those of passive polymers.’

Referee #3: In light of these concerns, it is imperative that the authors revisit their work to address the critiques outlined above. Additionally, based on recent findings, notably from Zheng et al. *Macromol. Rapid Commun.*, 2023, 44(1): 2200159, it is crucial to analyze the contribution of inter-chain forces to the modulus in order to substantiate claims regarding the influence of grip forces on the material properties.

Reply: We thank the referee for the thoughtful comments and for drawing our attention to the recent work of Zheng et al. [*Macromol. Rapid Commun.*, 2023, 44(1): 2200159]. We have carefully revisited our analysis in light of

Figure 9: **Time evolution of grip forces.** Total force on the first 150 beads of three test polymer chains as a function of bead number, showing the intensification of grip forces over time: (a) $tv/L = 0.037$, (b) $tv/L = 0.041$, and (c) $tv/L = 0.046$. The polymer has a length of $L = 1450\sigma$ and a Péclet number of $Pe = 4$. As time progresses, the grip forces exerted by neighboring polymers become stronger, leading to the stretching of the test polymer at key entanglement points. The insets depict one of the test polymers (green) and its primitive path (red), along with the primitive paths of neighboring polymers (blue), emphasizing the evolving topological interactions and constraints over time. Over the observed time interval, the contour length of the test polymer increases from 377σ to 389σ .

these critiques and have incorporated additional assessments to address the role of grip forces in influencing the material’s mechanical properties.

As part of our study, we had already performed a primitive path analysis based on the framework developed by Doi and Edwards and further validated by Everaers et al. [Science 303, 823 (2004)]. Our analysis confirms that the neighboring self-propelled polymers exert significant grip forces on the test polymer at each entanglement point, which leads to stretching of the polymer’s contour length [see Fig. 1(c) in the main text]. To visually support this, we have prepared a supplementary movie that illustrates this dynamic interaction [see Supplementary Movie].

In line with the referee’s suggestion, we have now extended our analysis to quantify the magnitude of the forces along the polymer contour. Specifically, we present data for a subsection of three test polymers (first 150 beads) in Fig. 9, showing the total force on each bead, $|\mathbf{f}_{tot}| = |\mathbf{F}_{WCA} + \mathbf{F}_{FENE} + \mathbf{F}_{ang} + \mathbf{F}_p|$, over time. Here, \mathbf{F}_{WCA} , \mathbf{F}_{FENE} , and \mathbf{F}_{ang} represent the forces due to the WCA, FENE, and angular potentials, respectively. At $tv/L = 0.037$, the total force fluctuates around $|\mathbf{f}_{tot}| \approx 50k_B T/\sigma$. As time progresses, grip forces from neighboring polymers (blue) intensify, increasing $|\mathbf{f}_{tot}|$ by a factor of 4 at $tv/L = 0.046$. These amplified forces stretch the primitive path (red) at key entanglement points, resulting in an increase in the contour length of the test polymer from 377σ to 389σ .

This new analysis provides a more detailed understanding of the role grip forces play in shaping the mechanical properties of the active polymer solutions. We believe that this deeper evaluation reinforces our original conclusions regarding the critical influence of grip forces on the material’s behavior.

’To investigate the nature of grip forces, we extended our analysis by quantifying the forces along the polymer contour. In Fig. 9, we present data for a subsection of the test polymer (first 150 beads), showing the total force on each bead, $|\mathbf{f}_{tot}| = |\mathbf{F}_{WCA} + \mathbf{F}_{FENE} + \mathbf{F}_{ang} + \mathbf{F}_p|$, over time. Here, \mathbf{F}_{WCA} , \mathbf{F}_{FENE} , and \mathbf{F}_{ang} represent the forces due to the WCA, FENE, and angular potentials, respectively. At $tv/L = 0.037$, the total force fluctuates around $|\mathbf{f}_{tot}| \approx 50k_B T/\sigma$. As time progresses, grip forces from neighboring polymers (blue) intensify, increasing $|\mathbf{f}_{tot}|$ by a factor of 4 at $tv/L = 0.046$. These amplified forces stretch the primitive path (red) at key entanglement points, resulting in an increase in the contour length of the test polymer from 377σ to 389σ .’

Referee #3: The manuscript presents a compelling exploration of polymer dynamics; however, there are significant concerns regarding the validity of the model employed. Notably, considering the relaxation time of the tube segment ($\tau_e \sim 10^3$) in the Kremer-Grest model, typical simulations (as reported in Nihon Reoraji Gakkaishi 2018, 46(5): 207-220; Macromolecules 2021, 54(6): 2811-2827; ACS Macro Lett. 2021, 10(12): 1517-1523) suggest that the monomer flow velocity should not exceed approximately 0.01. In contrast, the

Figure 10: Number of entanglement points (Z) as a function of the Péclet number (Pe) for (a) loosely entangled polymer solutions with length $L = 100\sigma$ and (b) highly entangled polymer solutions with length $L = 1450\sigma$. The plots illustrate how increasing activity (Pe) affects the number of entanglement points in both regimes.

Péclet number ($Pe = v\sigma/D_0 \sim v$) reported in the current work ranges from 4 to 24, implying that the monomer flow velocity (v) exceeds its thermal motion speed (~ 1). This leads to the conclusion that the polymer chains experience exceedingly high shear rates, rendering the foundational concepts of the tube model ineffective. I recommend that the authors reevaluate and revise their discussions throughout the manuscript in light of this discrepancy to ensure the accuracy and relevance of their findings.

Reply: We respectfully disagree with the referee’s concerns for several reasons.

First, the referee seems to have misunderstood the role of activity in our model. If $Pe \leq 0.1$, the system remains in the linear response regime, with polymers behaving as though they are undergoing passive Brownian motion. Our work explores the system beyond this regime, focusing on how adding an active component drives the system far from equilibrium. This exploration represents a significant advancement in understanding the interplay between entanglement and activity in polymer-like systems, with implications across various scales—from the cytoskeletons of cells to bacterial colonies and even bio-inspired engineering applications, such as soft robotic grippers that utilize active filaments for object manipulation.

Second, the referee appears to have misunderstood a key point regarding flow velocity. The claim that the flow velocity in the cited reference [Nihon Reoraji Gakkaishi 2018, 46(5): 207-220] should not exceed approximately 0.01 is incorrect. It seems likely that the referee intended to refer to the shear rate rather than the flow velocity. If we consider a shear rate of the order of $0.01\tau_0$, particles near the boundary would indeed experience a flow velocity of approximately $\dot{\gamma}y \simeq 1\sigma/\tau_0$ having a simulation box size of 100 [see Fig. 7(b)]. Notably, studies by Kröger and Hess [*Phys. Rev. Lett.* **85**, 1128 (2000)] report simulations where the shear rate exceeds $\dot{\gamma}/\tau_0 \simeq 1$, resulting in flow velocities ~ 100 that exceeds those in our simulations, where the Péclet number ($Pe = v\sigma/D_0 \sim v$) ranges from 4 to 24.

Third, the referee suggested studies [Nihon Reoraji Gakkaishi 2018, 46(5): 207-220] involving polymers with relatively few entanglement points per chain (approximately 2 for $L/\sigma = 100$) as shown in Fig. 10(a). In such cases, it is expected that under moderate deformation ($\dot{\gamma}$), these entanglement points may be lost, rendering the tube model ineffective. However, this is precisely why our study focuses on highly entangled systems, where each polymer chain initially has around 45 entanglement points [see Fig. 10(b)]. Even at the highest levels of activity, our simulations show that each polymer chain retains about 25 entanglement points [see Fig. 10(b)]. This issue has been discussed in our manuscript, where we emphasize the relevance of our findings to highly entangled systems.

We have revised the manuscript to clarify these points further and to ensure that our discussion accurately reflects the relevance and applicability of our model to the systems under study.

‘in the regime where $\tau_{\text{eff}} \gg \tau_e$ (i.e., $Pe \ll 1$), the polymers remain in the linear response regime, behaving similarly to equilibrium systems despite the influence of active forces. This comparison highlights the interplay between activity and entanglement effects, deepening our understanding of the dynamics that extend beyond those of passive polymers.’

'Looking ahead, it would be of great interest to investigate how these active entangled polymer solutions behave under deformation or shear [59-63], introducing another timescale (inverse shear rate) that could reveal nontrivial viscoelastic properties.'

'While self-propulsion is expected to reduce the number of entanglement points, potentially compromising the validity of the tube model, this effect is most pronounced in systems with relatively few entanglements per polymer chain (e.g., approximately 2 entanglements for $L/\sigma = 100$), as shown in Fig. 12(a). In such cases, even moderate activity can lead to the loss of these entanglements, rendering the tube model ineffective. This is precisely why our study focuses on highly entangled systems, where each polymer chain initially possesses around 45 entanglement points [see Fig. 12(b)]. Remarkably, even at the highest levels of activity, our simulations reveal that each chain retains approximately 25 entanglement points, demonstrating that significant entanglement persists despite the presence of active forces [see Fig. 12(b)].'

1. Reply to Referee #1

Referee #1: I thank the authors for the detailed replies to the concerns and questions from myself and the other two referees. The paper is well organized. Most of my concerns are resolved in the updated manuscript, except the relaxation modulus data shown in Figure 10(a). Based on the provided moduli results, the viscoelastic properties represented by storage and loss modulus should vary quite a lot after, for example changing the chain length from 725 to 2610. It will be great if the authors can show also the corresponding G' and G'' data by making a Fourier transfer of $G(t)$.

The paper should be published as it is after a minor revision.

We sincerely thank the Referee for their positive assessment of our manuscript and their recommendation that the paper “*should be published after a minor revision.*” We greatly appreciate the insightful suggestion to compute the linear viscoelastic properties, specifically the storage modulus $G'(\omega)$ and loss modulus $G''(\omega)$, by performing a Fourier transform of the stress relaxation modulus $G(t)$ [Rubinstein and Colby, Polymer Physics, Oxford University Press (2003)].

To address this, we have computed $G'(\omega)$ and $G''(\omega)$ from $G(t)$ using the standard relations:

$$G'(\omega) = \omega \int_0^{\infty} G(t) \sin(\omega t) dt, \quad G''(\omega) = \omega \int_0^{\infty} G(t) \cos(\omega t) dt. \quad (1)$$

Here, $G'(\omega)$ represents the storage modulus, which quantifies the system’s elastic response, and $G''(\omega)$ characterizes the viscous dissipation. The frequency-dependent shear moduli are ideally suited to distinguish elastic behavior (dominant at high frequencies, $G' > G''$) from viscous behavior (dominant at low frequencies, $G' < G''$) in polymer systems [see Fig. 1(a)].

We have incorporated the results in the revised manuscript and highlight the following key findings:

- 1. Crossover between G' and G'' at low frequencies:** The crossover frequency, where G' and G'' intersect, reflects the transition from viscous to elastic behavior and marks the effective disengagement time τ_{eff} [see Fig. 1(a)]. Our results confirm that $\tau_{\text{eff}} \sim L/v$, validating the key scaling predictions [see Fig. 1(b)].
- 2. Low-frequency scaling:** At low frequencies ($\omega \rightarrow 0$), $G'(\omega)$ scales as ω^2 . This scaling arises from the expansion $\sin(\omega t) \approx \omega t$, leading to $G'(\omega) \sim \omega^2 \int_0^{\infty} G(t)t dt \sim \omega^2 G_0 \tau_{\text{eff}}^2 \sim \omega^2 L^3/v$, where L is the polymer length and v is the self-propulsion speed [see Fig. 2(a)]. Similarly, $G''(\omega)$ scales linearly as $G''(\omega) \sim \omega \int_0^{\infty} G(t) dt \sim \omega G_0 \tau_{\text{eff}} L^2$, consistent with the series expansion $\cos(\omega t) \approx 1$ [see Fig. 2(b)].
- 3. Elastic plateau at intermediate frequencies:** At intermediate frequencies, $G'(\omega)$ displays a plateau, a hallmark of entangled polymer systems [see Fig. 2(a)]. The plateau modulus G_0 is extracted from the minimum in the loss-to-storage ratio, $\tan \delta = G''/G'$ [see Fig. 3(a)]. The results confirm the scaling $G_0 \sim L$ at a fixed Péclet number, as shown in Fig. 3(b).

These results provide a comprehensive characterization of the viscoelastic properties of active entangled polymer solutions and validate the scaling laws derived in our theoretical framework. The additional insights strengthen the manuscript by linking the stress relaxation modulus $G(t)$ to the frequency-dependent shear moduli $G'(\omega)$ and $G''(\omega)$, and by clarifying the dynamic behavior of active polymer systems.

We again thank the Referee for their thoughtful suggestion and have added a discussion in the main text and the Supplementary Material (SM).

‘To further characterize the viscoelastic properties of active entangled polymers, we compute the frequency-dependent storage modulus $G'(\omega)$ and loss modulus $G''(\omega)$ by applying a Fourier transform to the stress relaxation modulus $G(t)$ [42]. Our results reveal a well-defined crossover frequency ω_c , where $G'(\omega)$ and $G''(\omega)$ intersect, marking the transition from a viscous-dominated to an elastic-dominated regime. Notably, this crossover frequency scales as $\omega_c \sim 1/\tau_{\text{eff}}$, confirming that relaxation dynamics are governed by activity-driven disengagement [see Supplemental Material (SM) [49]].

‘... These results provide a comprehensive characterization of the viscoelastic properties of active entangled polymer solutions and validate the scaling laws derived in our theoretical framework.’

Figure 1: **Frequency-dependent viscoelastic properties and polymer disengagement time.** (a) Frequency-dependent storage modulus G' (empty symbols) and loss modulus G'' (open symbols) as functions of the frequency ω . The crossover point (indicated by markers) at which G' and G'' intersect reveals the timescale at which active polymers escape their confinement within entanglement tubes. (b) The extracted disengagement time τ_{eff} , plotted as a function of polymer length L , follows the scaling relation $\tau_{\text{eff}} \sim L/v$, where v is the self-propulsion speed.

Figure 2: **Frequency-dependent viscoelastic properties of active entangled polymer solutions.** (a) Storage modulus G' and (b) loss modulus G'' as functions of the frequency ω . At low frequencies, G' and G'' exhibit scaling behavior characteristic of viscous-dominated relaxation. The inset in (a) highlights the low-frequency prediction of the storage modulus $G'_0 = G'(\omega \rightarrow 0)$, while the inset in (b) shows the corresponding prediction for the loss modulus $G''_0 = G''(\omega \rightarrow 0)$. The results demonstrate the transition from viscous to elastic behavior, revealing the hallmark features of entangled active polymer systems.

Figure 3: **Extraction of the plateau modulus G_0 from the minimum of the loss-to-storage ratio.** (a) The minimum in $\tan \delta = G''/G'$ occurs at a frequency near the center of the elastic plateau. The storage modulus G' at this frequency provides a robust estimate of the plateau modulus G_0 . (b) The extracted G_0 scales linearly with polymer length L , confirming the prediction $G_0 \sim L$ at a fixed Péclet number.

Referee #1: One small concern: Does the dependence of zero shearing viscosity on active chain length, i.e $\eta \sim L^2$, work for short but entangled active chains (with length shorter 725 but long enough to entangle) ? Not a big change from $L=725$ to 2610 from the logarithmic perspective.

Reply: We thank the Referee for the question regarding the scaling behavior of the zero-shear viscosity $\eta_\infty \sim L^2$ for shorter but sufficiently entangled active chains. As shown in Fig. 4(c), our data confirm that this scaling holds robustly in the highly entangled regime where $L/\sigma \geq 350$.

We hope this clarification addresses the Referee's concern and we thank them again for the constructive comment.

Figure 4: **Fluidization of active solutions compared to their passive counterparts.** (a) Time-dependent viscosity for a wide range of Péclet numbers and polymer lengths L . (b) A data collapse is obtained by rescaling the viscosity by $\eta(t)\sigma^3/\zeta L^2$ and the time scale by tv/L . (c) Long-time viscosity η_∞ as a function of polymer length L extracted from simulations for a wide range of Péclet numbers. The black line indicates the scaling of $\eta_\infty \sim L^2$.

2. Reply to Referee #3

Referee #3: The stress relaxation modulus is the central physical quantity of this manuscript, and the significant conclusions drawn are closely linked to the results obtained from this modulus. However, calculating the stress relaxation modulus using the Green-Kubo formula requires specific assumptions and conditions to be satisfied. Unfortunately, the authors have not adequately demonstrated the validity of employing the Green-Kubo formula to calculate the system’s stress relaxation modulus under high Péclet numbers, as the Green-Kubo framework typically becomes invalid at elevated shear or stretching rates.

The authors reference the work of Han *et al.* (Nature Physics 17, 1260–1269, 2021), asserting that the Green-Kubo formula is applicable under high Péclet conditions. However, it is noteworthy that Han *et al.* clarify that an equilibrium-like Green–Kubo relation for the shear viscosity tensor holds near the steady state of any isotropic active fluid satisfying the following three conditions: (i) the activated and fluctuating degrees of freedom are statistically decoupled; (ii) the steady state is stable under small perturbations; (iii) the ensemble-averaged (microscopic) velocity-velocity correlations decay faster than r^{-D} (where D is the dimension of the system). Under high Péclet conditions, polymers experience stretching and orientation, resulting in a non-isotropic state, thus violating the isotropy assumption. Furthermore, unlike the approach of Han *et al.* (Nature Physics 17, 1260–1269, 2021), which incorporates a torque as an activated force on each monomer, the methodology in the current manuscript diverges significantly. Lastly, Han *et al.* focused on viscosity under small perturbations (linear response), suggesting that their definition of a “steady state” may not align with the definition utilized in this study

Reply:

We fully agree with the referee that the application of the Green-Kubo formalism deserves a more explicit justification. Below (and in the revised manuscript), we show in detail that the criteria for applying Green-Kubo relations far from equilibrium clearly apply to our work.

Figure 5: **Correlation between Brownian and self-propulsion forces.** The time evolution of the correlation between fluctuating Brownian forces and self-propulsion forces, $\langle \mathbf{F}_r \cdot \mathbf{F}_p \rangle$. The correlation remains close to zero on average, as indicated by the black dashed line, confirming the statistical independence of thermal fluctuations and active driving forces.

1. **Decoupling of activated and fluctuating degrees of freedom:** In our model, the fluctuating Brownian forces \mathbf{F}_r , which determine the thermal noise of the system, are Gaussian white noise and remain statistically uncorrelated with the self-propulsion forces \mathbf{F}_p (activated degrees of freedom) [see Fig. 5]. This statistical decoupling ensures that the dynamics of the activated forces are not influenced by thermal fluctuations.
2. **Stability of the steady state:** Han *et al.* (Nature Physics 17, 1260–1269, 2021) demonstrated that active systems perturbed slightly from their steady state relax back to this steady state rather than to equilibrium. Consistent with this principle, our active polymer system operates in a stable non-equilibrium steady state. Specifically, we observe that:
 - Stress fluctuations average to zero in the steady state [see Fig. 6].
 - The number of entanglements saturates after the disengagement time $\tau_{\text{eff}} = L/v$, where L is the polymer length and v is the self-propulsion velocity [see Fig. 7].

These observations indicate the stability of the steady state.

3. **Isotropy of the system:** We compute the radial distribution function (RDF) and confirm that our system remains isotropic, even at high Péclet numbers [see Fig. 8]. To understand the evolving orientational order, we have further calculated the nematic order parameter $P_2(n) = \frac{1}{2}(3\langle \cos^2(\theta_n) \rangle - 1)$, where $\langle \rangle$ indicates an ensemble average and θ_n is the angle between the z -axis and the vector \mathbf{R}_n between beads separated by n bonds. One expects that $P_2(N_p - 1) = 0$ for randomly oriented chains and $P_2(N_p - 1) = 1$ for perfectly aligned chains [O'Connor *et al.*, PRL 2019]. It turns out that our systems shows $P_2(N_p - 1) \approx 0$ [see Fig. 9(a)], which further validates that our system remains isotropic.

Here, we would like to emphasize that anisotropy in the RDF is a prerequisite for non-zero shear stress [Koumakis *et al.* PRL 2012, Svetlizky *et al.* PRL 2021]. In contrast, the isotropy in the RDF observed in our system ensures that shear stress fluctuations average to zero [see Fig. 6], further validating the assumptions of the Green-Kubo framework.

4. **Decay of the spatial velocity-velocity correlations:** To confirm the final condition, we calculate the velocity autocorrelation functions [see Fig. 9(b)]. Analyzing the spatial velocity correlations we find that they decay beyond the polymer (monomer) diameter, consistent with a decay faster than r^{-3} in three dimensions. This result satisfies the condition that velocity correlations must decay sufficiently rapidly, as outlined by Han *et al.* (Nature Physics 17, 1260–1269, 2021).

Finally, we note that Green-Kubo relations have been successfully applied to other active systems, such as active Brownian particles (ABPs), where transport coefficients are derived using non-equilibrium steady-state correlation functions as the reference state. Furthermore, a Green-Kubo relation has been developed to explore the odd diffusivity and the odd viscosity in non-equilibrium set-ups [Hargus *et al.*, PRL 2021, Ghimenti *et al.*, PRL 2023, Matus *et al.*, PRE 2024].

We hope this detailed clarification resolves the Referee’s concerns and clarifies the validity of our approach.

Figure 6: **Time evolution of shear stress for different Péclet numbers.** Time-dependent shear stress $\sigma_{xy}(t)$ for different Péclet numbers: (a) $Pe = 1$, (b) $Pe = 4$, and (c) $Pe = 8$. The figure illustrates how shear stress evolves over time under varying Péclet numbers.

Our verification of key conditions, including the statistical independence of thermal and active forces, the stability of the steady state, and the isotropy of the system, and confirmation that spatial velocity correlations decay sufficiently fast, ensures the applicability of the formalism also for our work [see SM [49]]. Thus, the Green-Kubo relation offers access to the time-dependent viscosity of our entangled system via

Referee #3: The authors claim that the effective active timescale, $\tau_{\text{eff}} = L/v$, is typically shorter than τ_e . However, on timescales shorter than τ_e , the tube model clearly becomes ineffective, diluting the strength of any comparisons made with the tube model within the manuscript.

Reply: We sincerely appreciate the Referee’s insightful comment and the opportunity to clarify this key aspect of our work. Our study introduces a fundamentally new theoretical framework—the *active tube model*—which differs from the classical equilibrium tube model used in passive polymer systems. In passive systems, the characteristic timescale $\tau_e \sim N_e^2$ (where N_e is the entanglement length) governs the relaxation of polymers within a confining

Figure 7: **Time evolution and steady-state behavior of entanglement points across Péclet numbers.** (a) Time evolution of the number of entanglement points, Z , for various Péclet numbers (Pe) at a fixed polymer length $L = 1450\sigma$. (b) Number of entanglement points at the steady state as a function of Pe , illustrating how increasing activity affects polymer entanglement points.

Figure 8: **Isotropy of the radial distribution function across varying Péclet numbers.** The radial distribution function $g_{xy}(r, \theta)$ in the xy -plane for various Péclet numbers (Pe) shows that the system remains isotropic, even with increasing activity.

Figure 9: **Orientalional order and velocity correlations in active polymers across varying Péclet numbers.** (a) The orientational order parameter $P_2(N_p - 1)$, where N_p is the number of monomers per chain, as a function of the Péclet number Pe . The data show how the orientational order evolves with increasing activity, reflecting the degree of alignment of polymer chains under active driving. Despite increasing Pe , the system maintains a relatively low orientational order (≈ 0), indicating that alignment does not develop even at high activity levels. (b) The spatial velocity correlation $\langle \mathbf{v}_i(r) \cdot \mathbf{v}_i(0) \rangle$, where $\mathbf{v}_i = d\mathbf{x}_i/dt$, as a function of Péclet number Pe . The results show that the velocity correlations decay rapidly, vanishing once the displacement distance r exceeds the monomer diameter. This rapid decay indicates that the system lacks long-range velocity correlations, even at high activity levels.

Figure 10: **Dynamic rearrangement of the entangled environment in active polymers.** In the highly entangled regime, a self-propelled polymer is confined within an effective tube formed by the surrounding self-propelled polymers. Over time, the polymer escapes its initial tube and becomes confined within a newly formed tube, reflecting the dynamic rearrangement of the entangled environment driven by activity.

tube. However, in our active system, self-propulsion drives each polymer to exit its confining tube on a much shorter timescale $\tau_{\text{eff}} = L/v$. This timescale arises because self-propulsion induces a persistent sliding motion of the polymer, which prevails over random thermal fluctuations and enables the polymer to escape its tube more quickly.

It is important to note that, while the effective tube turnover is rapid (on the timescale τ_{eff}), the system remains highly entangled. Specifically: (a) Upon exiting its tube, an active polymer immediately enters a new tube formed by the surrounding polymers [see Fig. 10]. (b) The tubes themselves are dynamic, continually breaking and reforming on the same timescale $\tau_{\text{eff}} = L/v$.

Our analysis reveals that the system retains a well-entangled structure, with the number of entanglements exceeding 20 for highly entangled active polymers [see Fig. 11(b)]. This observation suggests that the entanglement framework remains robust, even in the presence of rapid self-propelled dynamics. Hence, we use a new model (which could be called an active tube model), not the equilibrium tube model.

To further clarify this point, we have expanded the **Methods** section to elaborate on the implications of τ_{eff} for entanglement dynamics and the validity of our framework.

We hope this response effectively addresses the Referee’s concern and highlights the novelty of our findings in the context of active polymer dynamics.

Our study introduces a fundamentally new theoretical framework—the active tube model—which differs from the classical equilibrium tube model used in passive polymer systems. In passive systems, the characteristic timescale $\tau_e \sim N_e^2$ (where N_e is the entanglement length) governs the relaxation of polymers within a confining tube. However, in our active system, self-propulsion drives each polymer to exit its confining tube on a much shorter timescale $\tau_{\text{eff}} = L/v$. This timescale arises because self-propulsion induces a persistent sliding motion of the polymer, which prevails over random thermal fluctuations and enables the polymer to escape its tube more quickly.

It is important to note that, while the effective tube turnover is rapid (on the timescale τ_{eff}), the system remains highly entangled. Specifically: (a) Upon exiting its tube, an active polymer immediately enters a new tube formed by the surrounding polymers [see Fig. 10]. (b) The tubes themselves are dynamic, continually breaking and reforming on the same timescale $\tau_{\text{eff}} = L/v$.

Our analysis reveals that the system retains a well-entangled structure, with the number of entanglements exceeding 20 for highly entangled active polymers [see Fig. 11(b)]. This observation suggests that the entanglement framework remains robust, even in the presence of rapid self-propelled dynamics. Hence, we use a new model (which could be called an active tube model), not the equilibrium tube model.

Referee #3: In the revised manuscript, the authors explore the viscoelastic properties of less entangled systems (with $L = 25$) and provide Figure 9a to account for the factor of increased monomeric friction. Nonetheless, Ianniruberto et al. (Macromolecules 48, 6306-6312, 2015) have indicated that the friction coefficient of monomers decreases for polymers subjected to orientation and stretching. I suggest that the authors incorporate a focused discussion addressing this point.

Figure 11: **Entanglement points as a function of activity in low and high entangled polymer.** Number of entanglement points (Z) as a function of the Péclet number (Pe) for (a) loosely entangled polymer solutions with length $L = 100\sigma$ and (b) highly entangled polymer solutions with length $L = 1450\sigma$. The plots illustrate how increasing activity (Pe) affects the number of entanglement points in both regimes.

Reply: We thank the Referee for raising this insightful point. We observe that the short-time monomeric friction and the terminal relaxation times in our active polymer solutions differ significantly from those of passive systems. In our study, the increased monomeric friction arises due to enhanced fluctuations caused by self-propulsion forces, where each polymer actively propels in random directions. These fluctuations generate an effective resistance that increases monomer friction compared to passive systems.

In contrast, the work by Ianniruberto *et al.* (Macromolecules 48, 6306–6312, 2015) focuses on passive polymer solutions under uniaxial extension. Under such conditions, polymer chains align with the flow direction, reducing the monomer friction coefficient due to the reduced resistance. The key distinction here is the driving mechanism: in uniaxially deformed passive systems, alignment and stretching reduce friction, whereas in our active system, random self-propulsion leads to enhanced friction through persistent fluctuations.

We acknowledge that exploring the behavior of active polymer solutions under uniaxial tension would indeed be fascinating. Such a study could provide further insight into the interplay between alignment, activity, and monomeric friction. We highlight this as an exciting direction for future research.

We hope this clarifies the distinction between our findings and the results presented in Ianniruberto *et al.* (Macromolecules 48, 6306–6312, 2015), and we appreciate the Referee’s suggestion.

‘In contrast to our findings, previous work [74] focuses on passive polymer solutions under uniaxial extension. Under such conditions, polymer chains align with the flow direction, reducing the monomer friction coefficient due to the reduced resistance. The key distinction here is the driving mechanism: in uniaxially deformed passive systems, alignment and stretching reduce friction, whereas in our active system, random self-propulsion leads to enhanced friction through persistent fluctuations.’

Referee #3: In my previous review, I requested that the authors provide the friction coefficient relevant to their model; however, this essential parameter remains unaddressed. Additionally, the authors should specify the maximum bond length associated with the FENE potential.

Reply: We thank the Referee for their valuable suggestions and the opportunity to clarify these points. In our simulations, the friction coefficient ξ is determined from the Stokes-Einstein relation, $\xi = k_B T / D_0$, where D_0 is the short-time diffusion coefficient. We use the characteristic time unit $\tau_0 = \sigma^2 / D_0$, and in dimensionless units, the friction coefficient is set to $\xi = 1$. Regarding the FENE potential, we set the maximum bond length to $R_0 = 1.5\sigma$ in our simulations.

‘In our simulations, the friction coefficient is set to $\xi = k_B T / D_0 = 1$. For the FENE potential, we define the maximum bond length as $R_0 = 1.5\sigma$.’

Referee #3: To maintain a constant system temperature, it is recommended that the velocity not exceed approximately 0.01. Kröger and Hess (Phys. Rev. Lett. 85, 1128, 2000) employed a velocity rescaling method to achieve high flow velocities. The authors need to verify whether there has been any change in the system’s

temperature, and assess potential issues related to excessively high local shear rates.

Reply: We thank the Referee for raising this issue and appreciate the opportunity to clarify our approach. Kröger and Hess [Phys. Rev. Lett. **85**, 1128 (2000)] performed molecular dynamics simulations under shear flow, where velocities are direct observables, and energy is continuously pumped into the system to obtain a shear profile $v = \dot{\gamma}y$ (with $\dot{\gamma}$ as the shear rate and y as the spatial coordinate). In such cases, temperature control through a thermostat, such as velocity rescaling, is essential to prevent temperature increases caused by the added energy.

However, in our study, we performed simulations in the overdamped regime [Doi and Edwards, *Theory of Polymer Dynamics*, Oxford University Press (1986)], where inertia is negligible compared to friction ($m/\xi \ll 1$). Under these conditions, the equation of motion simplifies to:

$$\xi \frac{d\mathbf{r}_i}{dt} = -\nabla_i U + \mathbf{F}_{p,i} + \mathbf{F}_{r,i}, \quad (2)$$

where $\mathbf{F}_{r,i}$ represent the random forces on bead i that are normally distributed and governed by the fluctuation-dissipation theorem, ensuring thermal equilibrium with the bath at temperature $k_B T$. Importantly, in this overdamped limit, momentum dissipates instantaneously, and the system inherently remains at the bath temperature without requiring an explicit thermostat. This behavior has also been demonstrated in recent studies, such as Wiese [Phys. Rev. Lett. **131**, 17830 (2023)], where active Brownian particles under shear flow maintained the bath temperature across a broad range of shear rates (10^{-7} to 10).

We have clarified these points in the revised manuscript to address the Referee’s concern regarding thermal control and potential shear-related artifacts.

‘In the overdamped regime, momentum dissipates immediately, ensuring that the system consistently remains at the bath temperature, $k_B T$, making an explicit thermostat unnecessary [32].’

Referee #3: Furthermore, in the Abstract, it would be beneficial to provide specific numerical values or examples regarding the viscosity changes induced by activity. Additionally, while the prospects of the study are mentioned, the authors should clarify future research directions or specific application cases.

Reply: In response to the request for quantitative details, we predict that activity significantly fluidizes the polymer suspension, leading to a long-time viscosity scaling as $\eta \sim L^2$, in contrast to the equilibrium scaling of $\eta \sim L^3$. Consequently, the long-time viscosity is reduced by a factor proportional to the polymer length, highlighting the significant impact of activity.

Regarding the prospects of our study, we emphasize that activity-enhanced elasticity, as revealed in our work, is a critical property for collective life forms. This feature can provide individuals with resistance to environmental stresses by tuning their activity [Deblais et al. PRL 2020, Day et al. PRX 2023]. The implications of our findings extend to numerous technological and biological applications, such as:

- **Advanced material design:** Our results pave the way for developing materials composed of microscale activated nanotubes [Bacanu et al. Nat. Nanotechnology (2023)], synthetic polymer chains [Yang et al. PNAS (2020)], rigid helical filaments [Schamel et al. ACS Nano (2014), Yardimci et al. PNAS (2023)], or soft shape-changing actuators [Jones et al. Nature (2021) and Becker et al. PNAS (2022)].
- **Entangled living systems:** Our study offers insights into predicting the elasticity of growing entangled living systems, such as bacterial colonies [van Dissel et al. Adv. Appl. Microbiol. (2014) and Day et al. PRX (2024)]. For instance, as filamentous bacterial strands grow, we anticipate that the plateau of the stress autocorrelation function—and consequently the elasticity—will increase over time.

‘resulting in a viscosity reduction by a factor proportional to the polymer’s own length.’

‘This intricate feature also lays the foundation for numerous technological applications, involving new materials composed, for example, of microscale activated nanotubes [56], synthetic polymer chains [57], rigid helical filaments [58,59], or soft shape-changing actuators [18,19]. Furthermore, our findings hold significant promise in predicting the elasticity of growing entangled living systems, such as bacterial colonies [14,16], as our simulation results across various polymer lengths can effectively map onto growing matter. In particular, we anticipate that the plateau of the stress autocorrelation function, and hence the elasticity, increases over time due to the growth of filamentous bacterial strands.’

Referee #3: The simplified model employed in this study may not adequately represent the complexities of actual active entangled polymer solutions, which typically involve additional variables and interactions. I recommend that the authors discuss approaches to validate the accuracy of the simplified model in real active entangled polymer systems.

Reply: We acknowledge that our current study employs a minimalistic model to explore the fundamental physics of active entangled polymer solutions. The simplicity of our approach allows us to isolate the role of activity and entanglement in determining the emergent viscoelastic behavior, serving as a foundation for guiding future experiments and more complex models. To validate our findings in real active polymer systems, we suggest two potential directions:

- **Collective biological systems:** Our minimal model provides a framework to understand the collective behavior of life-like systems, such as *Tubifex* worms, which form entangled active clusters and respond collectively to external stimuli. Future experimental comparisons with such systems could validate the predicted fluidization and elasticity effects.
- **Entangled biopolymer solutions:** Real active polymer systems, such as actin filament networks, involve additional complexities, including dynamic crosslinkers that bind and detach from filaments. These interactions could influence the structural and mechanical properties of the network. Thus, future work should incorporate such dynamic crosslinking effects, along with more detailed interactions, into our model to bridge the gap between minimalistic simulations and experimental systems.

We believe that our minimalistic approach captures the essential physics and sets the stage for exploring additional complexities, such as crosslinker dynamics, polydisperse polymers, and filament flexibility in more realistic settings. We have clarified this point in the revised manuscript.

'The activity-enhanced elasticity elucidated in our study is a critical property for collective life forms, which could provide individuals resistance to environmental stresses [16,17,51-55] via tuning their own activity. This intricate feature also lays the foundation for numerous technological applications, involving new materials composed, for example, of microscale activated nanotubes [56], synthetic polymer chains [57], rigid helical filaments [58,59], or soft shape-changing actuators [18,19]. Furthermore, our findings hold significant promise in predicting the elasticity of growing entangled living systems, such as bacterial colonies [14,16], as our simulation results across various polymer lengths can effectively map onto growing matter.'

'Finally, real active polymer systems, such as actin filament networks [65], involve additional complexities, including dynamic crosslinkers that bind and detach from filaments. These interactions could influence the structural and mechanical properties of the network. Thus, future work should incorporate such dynamic crosslinking effects, along with more detailed interactions, into our model to bridge the gap between simplified simulations and experimental systems.'

Referee #3: While the conclusions present a theoretical model and associated inferences, they lack direct comparisons and validations with experimental data. The reliability of these conclusions must be supported by experimental evidence to elevate their credibility.

Reply: Polymer physics has long benefited from theoretical predictions that have successfully guided and inspired experimental studies. While the field of active polymers is still emerging, recent experimental efforts, such as those by Deblais *et al.* [PRL, 2020], have begun to explore these systems. However, a comprehensive theoretical framework to guide such experiments remains lacking.

The pioneering work of Deblais *et al.* provides an experimental investigation of zero-shear viscosity in active entangled systems. Their study focused on *Tubifex* worms, which typically have lengths of 10–30 mm and widths of 0.2–0.4 mm, exhibiting active motion in water. By adding 5% alcohol, they were able to systematically tune the worms' activity, making them more passive. Their rheological measurements demonstrated that the zero-shear viscosity ($\dot{\gamma} \rightarrow 0$) decreases with increasing activity, qualitatively supporting our theoretical predictions.

A direct validation of our scaling predictions could be achieved by further experimental studies utilizing controlled worm lengths. Such investigations would provide a quantitative comparison and strengthen the connection between theoretical models and experimental observations in active polymer systems.

'Our work represents a minimal model for understanding the elasticity of active entangled polymer solutions, revealing a novel scaling law for elasticity, $G_0 \sim L$, for a fixed Péclet number. This contrasts with the conventional passive entangled polymer solutions, where elasticity remains constant regardless of polymer lengths [22].'

'Additionally, a direct validation of our scaling predictions could be achieved by further experimental studies [51] utilizing controlled worm lengths.'

1. Reply to Referee #3

Referee #3: The manuscript addresses a significant topic regarding the giant activity-induced elasticity in entangled polymer solutions. The revisions made to the paper have largely resolved my previous concerns. The manuscript is now suitable for publication after a minor revision.

(1) The authors should clarify that the simulation initiates the calculation of $G(t)$ when $t > L/v$ i.e., when the system enters a new steady state. This point should be explicitly mentioned to avoid any confusion regarding the timing of the calculations.

Reply: We sincerely thank the Referee for recommending our manuscript for publication in Nature Communications. We also appreciate the helpful suggestion and have now explicitly clarified in the manuscript that the stress autocorrelation function $G(t)$ is computed in the steady state.

‘The viscoelastic properties of polymer solutions are encoded in the stress autocorrelation function

$$G(t) = \frac{V}{3k_B T} \sum_{\alpha \neq \beta} \langle \sigma_{\alpha\beta}(t) \sigma_{\alpha\beta}(0) \rangle, \quad (1)$$

where the sum runs over all off-diagonal components of the stress tensor $\sigma_{\alpha\beta}$ (i.e., $\sigma_{xy}, \sigma_{yz}, \sigma_{xz}$), and $\langle \dots \rangle$ denotes the ensemble average, evaluated in the steady state ($t > L/v$).

Referee #3: (2) In discussing Fig. 3a, the authors refer to Fig. 1b, but the time scale indicated in Fig. 1b ($t < v/L$) corresponds to a non-equilibrium regime where the Green-Kubo formalism is not applicable. To prevent potential misunderstanding, I recommend modifying Fig. 1b to clearly reflect the correct time scale that corresponds to the equilibrium regime discussed.

Reply: We thank the Referee for this helpful comment. To eliminate potential confusion regarding the applicability of the Green-Kubo formalism, we have revised Fig. 1b accordingly. Specifically, we have updated the label from ‘ $0 < tv/L < 1$ ’ to ‘ $tv/L > 1$ ’ to clearly indicate that the stress autocorrelation function is computed only in the steady-state regime. Please see the updated Fig. 1 below.

Referee #3: (3) The authors should ensure that all parameters in the captions of the Supplemental Materials are clearly stated. Specifically, the N_p value in Fig. S4 and the Peclet number Pe in Fig. S5 should be explicitly mentioned to avoid any ambiguity.

Reply: We thank the Referee for this valuable suggestion. We have now explicitly stated the value $N_p = 725$ in the caption of Fig. 2 in the Supplemental Information (SI). Additionally, we have clearly indicated the Péclet number $Pe = 16$ in the captions of Figs. 3, 4, and 5 in the SI to ensure clarity.

Figure 1: **Exemplars of entangled active matter across scales and illustration of the physical mechanism driving their material properties.** (a) (*Top left panel*) 3D segmentation of actin filaments in native podosomes, scale bar 200 nm. Color code indicates their orientation. From Ref. [28]. (*Top right panel*) Growing community of snowflake yeast, scale bar 50 μm . From Ref. [29]. (*Bottom left panel*) Synthetic active filaments change curvature upon pneumatic actuation to entangle and capture different objects. From Ref. [19]. (*Bottom right panel*) Entangled blob of California blackworms, scale bar 3 mm. From Ref. [17]. At the center a simulation snapshot of entangled, flexible polymers (each polymer has its own color) is shown. (b) 3D illustration depicting the primitive path of a test polymer (red line) confined within an effective tube formed by surrounding self-propelled polymers at various times t . In the equilibrium state ($t = 0$), a combination of strong entanglement points (A, C, and D) and weak entanglement points (B) coexists, with strong entanglements distinguished by the presence of hairpin structures. Due to the activity, before reaching the steady state $t \gtrsim L/v$, the number of strong entanglement points increases (as shown by the yellow polymer wrapping around the red polymer at point B), leading to the elongation of the primitive path. The direction of self-propulsion is indicated by colored arrows, while the distance between successive entanglement points defines the entanglement length N_e . (c) Contour length of the primitive path L_{pp} , normalized by the equilibrium primitive path L_{pp}^0 as a function of time for different polymer lengths L and fixed Péclet number $Pe = 8$. Time is rescaled by the ratio of polymer length to self-propulsion velocity L/v . (d) Number of entanglement points Z , normalized by the number of entanglement points Z^0 for $Pe = 0$, as a function of time.

Figure 2: **Orientational order and velocity correlations in active polymers across varying Péclet numbers.** (a) The orientational order parameter $P_2(N_p - 1)$, where $N_p = 725$ is the number of monomers per chain, as a function of the Péclet number Pe . The data show how the orientational order evolves with increasing activity, reflecting the degree of alignment of polymer chains under active driving. Despite increasing Pe , the system maintains a relatively low orientational order (≈ 0), indicating that alignment does not develop even at high activity levels. (b) The spatial velocity correlation $\langle \mathbf{v}_i(r) \cdot \mathbf{v}_i(0) \rangle$, where $\mathbf{v}_i = d\mathbf{x}_i/dt$, as a function of Péclet number Pe . The results show that the velocity correlations decay rapidly, vanishing once the displacement distance r exceeds the monomer diameter. This rapid decay indicates that the system lacks long-range velocity correlations, even at high activity levels.

Figure 3: **Frequency-dependent viscoelastic properties and polymer disengagement time at a fixed Péclet number, $Pe=16$.** (a) Frequency-dependent storage modulus G' (empty symbols) and loss modulus G'' (open symbols) as functions of frequency ω . The crossover point (indicated by markers) at which G' and G'' intersect reveals the timescale at which active polymers escape their confinement within entanglement tubes. (b) The extracted disengagement time τ_{eff} , plotted as a function of polymer length L , follows the scaling relation $\tau_{\text{eff}} \sim L/v$, where v is the self-propulsion speed.

Figure 4: **Frequency-dependent viscoelastic properties of active entangled polymer solutions at a fixed Péclet number, $Pe=16$.** (a) Storage modulus G' and (b) loss modulus G'' as functions of frequency ω . At low frequencies, G' and G'' exhibit scaling behavior characteristic of viscous-dominated relaxation. The inset in (a) highlights the low-frequency prediction of the storage modulus $G'_0 = G'(\omega \rightarrow 0)$, while the inset in (b) shows the corresponding prediction for the loss modulus $G''_0 = G''(\omega \rightarrow 0)$. The results demonstrate the transition from viscous to elastic behavior, revealing the hallmark features of entangled active polymer systems.

Figure 5: **Extraction of the plateau modulus G_0 from the minimum of the loss-to-storage ratio at a fixed Péclet number, $Pe=16$.** (a) The minimum in $\tan \delta = G''/G'$ occurs at a frequency near the center of the elastic plateau. The storage modulus G' at this frequency provides a robust estimate of the plateau modulus G_0 . (b) The extracted G_0 scales linearly with polymer length L , confirming the prediction $G_0 \sim L$ at a fixed Péclet number.